# miR-155 harnesses Phf19 to potentiate cancer immunotherapy through epigenetic reprogramming of CD8+ T cell fate

Yun Ji [1,10,11], Jessica Fioravanti [1,11], Wei Zhu[2,11], Hongjun Wang[3], Tuoqi Wu[4], Jinhui Hu[1], Neal E. Lacey [1], Sanjivan Gautam[1], John B. Le Gall[1], Xia Yang[1], James D. Hocker[1], Thelma M. Escobar[5], Shan He[6], Stefania Dell'Orso [3], Nga V. Hawk[1], Veena Kapoor[1], William G. Telford[1], Luciano Di Croce [7,8,9], Stefan A. Muljo[5], Yi Zhang[6], Vittorio Sartorelli[3] & Luca Gattinoni [1]

T cell senescence and exhaustion are major barriers to successful cancer immunotherapy. Here we show that miR-155 increases CD8+ T cell antitumor function by restraining T cell senescence and functional exhaustion through epigenetic silencing of drivers of terminal differentiation. miR-155 enhances Polycomb repressor complex 2 (PRC2) activity indirectly by promoting the expression of the PRC2-associated factor Phf19 through downregulation of the Akt inhibitor, Ship1. Phf19 orchestrates a transcriptional program extensively shared with miR-155 to restrain T cell senescence and sustain CD8+ T cell antitumor responses. These effects rely on Phf19 histone-binding capacity, which is critical for the recruitment of PRC2 to the target chromatin. These findings establish the miR-155–Phf19–PRC2 as a pivotal axis regulating CD8+ T cell differentiation, thereby paving new ways for potentiating cancer immunotherapy through epigenetic reprogramming of CD8+ T cell fate.

[1] Experimental Transplantation and Immunology Branch, Center for Cancer Research, National Cancer Institute, National Institutes of Health, Bethesda, MD 20892, USA. [2] Department of Bioinformatics, Inova Translational Medicine Institute, Fairfax, VA 22031, USA. [3] Laboratory of Muscle Stem Cells and Gene Regulation, National Institute of Arthritis, Musculoskeletal and Skin Diseases, National Institutes of Health, Bethesda, MD 20892, USA. [4] National Human Genome Research Institute, National Institutes of Health, Bethesda, MD 20892, USA. [5] Laboratory of Immunology, National Institute of Allergy and Infectious Diseases, National Institutes of Health, Bethesda, MD 20892, USA. [6] Fels Institute for Cancer Research and Molecular Biology, Temple University, Philadelphia, PA 19140, USA. [7] Centre for Genomic Regulation (CRG), The Barcelona Institute of Science and Technology, Dr. Aiguader 88, 08003 Barcelona, Spain. [8] Universitat Pompeu Fabra (UPF), Barcelona 08003, Spain. [9] ICREA, Pg. Lluis Companys 23, 08010 Barcelona, Spain. [10] Present address: Cellular Biomedicine Group (CBMG), Gaithersburg, MD 20877, USA. [11] These authors contributed equally: Yun Ji, Jessica Fioravanti, Wei Zhu. Correspondence and requests for materials should be addressed to Y.J. (email: yji365@gmail.com) or to L.G. (email: gattinol@mail.nih.gov)

T cell senescence and exhaustion are major hurdles to successful cancer immunotherapy and treatment of chronic infectious disease[1,2]. Extensive effort has been devoted to boosting T cell immunity by either preventing or reverting these dysfunctional cellular states. These approaches include intrinsic modulation of pivotal transcription factors (TF)[3–5] and metabolic pathways[6] that regulate T cell differentiation and extrinsic blockade of inhibitory receptors[7] that enforce T cell functional exhaustion.

Small non-coding microRNA (miRNA) are a particularly attractive class of molecules that can be employed to efficiently modulate T cell differentiation and function[8]. Due to their ability to post-transcriptionally silence multiple molecules involved in essential developmental pathways, miRNAs act as potent regulators of cell fate commitment[9]. Emerging evidence has revealed the existence of a miRNA-epigenetic regulatory network that enables miRNAs to control cell fate through modulation of the epigenetic machinery[10]. For instance, specific miRNAs have been shown to influence the epigenetic make-up by directly targeting DNA methyltransferases[11], histone deacetylases[12], and Polycomb-group proteins[13,14].

We have recently demonstrated that miR-155, a miRNA essential to mounting productive pathogen-specific and tumor-specific CD8$^+$ T cell responses[15–17], could be harnessed to enhance T cell survival, restrain functional exhaustion, and augment antitumor immunity[18]. These functional advantages have been linked to miR-155 enhancement of CD8$^+$ T cell responsiveness to homeostatic cytokines via targeting of several inhibitors of cytokine signaling, including Socs1, Ship1, and Ptpn2[15,18,19]. Additionally, miR-155 has been proposed to regulate CD8$^+$ T cell sensitivity to type 1 interferon during anti-viral immune responses[16].

Here, we show a previously uncharacterized interplay between miR-155 and the epigenetic silencing complex, Polycomb repressor complex 2 (PRC2). We demonstrate that miR-155 restrains T cell senescence by epigenetically inhibiting key TFs driving terminal differentiation and exhaustion. Rather than directly modulating the epigenetic apparatus, miR-155 indirectly promotes the expression of Phf19, the mammalian ortholog of *D. melanogaster* Polycomb-like protein (Pcl), via pAKT to enhance PRC2 function. These findings reveal a new miRNA-epigenetic circuitry for guiding CD8$^+$ T cell fate decisions, which can be leveraged therapeutically to prevent terminal differentiation and exhaustion.

## Results

### miR-155 epigenetically silences CD8$^+$ T cell differentiation.

We previously showed in a melanoma model of adoptive T cell therapy that overexpression of miR-155 in CD8$^+$ T cells results in increased responsiveness to endogenous homeostatic cytokines, augmented engraftment, sustained cytokine production, and enhanced antitumor function[18]. To gain deeper insight into the molecular mechanisms underlying miR-155 activity, we sought to ascertain the gene expression profile of CD8$^+$ T cells overexpressing miR-155. We isolated pmel-1 CD8$^+$ T cells (which recognize the shared melanoma-melanocyte differentiation antigen gp100) transduced with miR-155 or a control vector 5 days after transfer into recipient mice infected with a recombinant strain of vaccinia virus encoding the cognate antigen gp100 (gp100-VV) and performed a massively parallel RNA-seq. Strikingly, Gene Set Enrichment Analyses (GSEA) revealed that eight of the 15 top-ranked enrichment sets were related to PRC2 activity in stem cells and progenitor cells (Supplementary Data 1). Specifically, miR-155-overexpressing cells showed reduced expression of genes silenced by PRC2 in mouse and human

embryonic stem cells (ESC) and progenitors[20,21] (Fig. 1a, Supplementary Fig. 1a and Supplementary Data 2), suggesting that miR-155 may promote PRC2 function in CD8$^+$ T cells. Corroborating these observations, we found that miR-155 overexpression significantly modulated the expression levels of PRC2 core complex members, PRC2 cofactors, and demethylases of trimethylated lysine 27 on histone H3 (H3K27me3) in CD8$^+$ T cells (Fig. 1b and Supplementary Fig. 1b).

Because PRC2 is known to maintain stem cell identity by restraining the expression of lineage-determining genes[22], we speculated that miR-155 inhibits T cell effector differentiation by modulating PRC2 activity. To test this hypothesis, we adoptively transferred pmel-1 cells overexpressing miR-155 or a control scramble miR (Ctrl-miR) into wild-type hosts and evaluated the development of terminally differentiated, KLRG1$^+$CD62L$^-$ effector (T$_E$) cells[2] in response to gp100-VV infection. As previously shown[18], CD8$^+$ T cells overexpressing miR-155 expanded and survived better than controls (Fig. 1c and Supplementary Fig. 2). Remarkably, despite the extensive T cell expansion, the frequency of terminally differentiated cells was significantly reduced in miR-155-overexpressing cells at all time points analyzed (Fig. 1d, e). A similar observation was recently reported in a mouse model of chronic viral infection[23]. To strengthen these observations, we measured the mRNA of inhibitory receptors, cytotoxic molecules, and chemokines known to be highly expressed in terminally differentiated CD8$^+$ T cells[24–26]. Consistently, we found that miR-155-overexpressing cells isolated at peak immune response expressed lower amount of senescence markers (Supplementary Fig. 3). To assess whether physiologic miR-155 levels also inhibit T cell differentiation, we evaluated the generation of T$_E$ cells in response to gp100-VV infection following adoptive transfer of CFSE-labeled wild-type and miR-155-deficient cells engineered to express the pmel-1 TCR. As we previously showed[15], *Mir155$^{-/-}$* cells demonstrated an expansion defect (Supplementary Fig. 4a–c). Even with defective expansion, miR-155-deficient cells generated similar frequencies of T$_E$ cells (Supplementary Fig. 4d, e). However, when we normalized the impact of cell expansion on T cell differentiation by measuring the frequency of T$_E$ cells in each cell division (Supplementary Fig. 4c), we observed that *Mir155$^{-/-}$* cells underwent expedited terminal differentiation (Fig. 1f, g). Altogether these results indicate that both physiologic and heightened miR-155 restrain CD8$^+$ T cell terminal differentiation.

To determine whether miR-155 limited T cell senescence through chromatin remodeling via PRC2 activity, we performed a genome-wide mapping of H3K27me3 in non-terminally differentiated KLRG1$^-$ cells overexpressing miR-155 or Ctrl-miR. We found that miR-155 overexpression globally enhanced the deposition of H3K27me3. About 1900 genes were uniquely marked by H3K27me3 in miR-155 overexpressing cells, compared to only 194 genes in Ctrl-miR-overexpressing cells (Supplementary Data 3). Next, we focused our analysis on a number of master regulators of terminal differentiation and exhaustion. We did not observe any H3K27me3 marks at the promoters of *Batf*[27], *Irf4*[28], and *Tbx21*[29] indicating that these TFs were not influenced by PRC2 activity at this stage of CD8$^+$ T cell differentiation (Supplementary Fig. 5a). Notably *Id2*[30], *Eomes*[31], *Prdm1*[32–34], *Zeb2*[35,36], *Maf*[26], and *Nr4a2*[26,37] were heavily marked with H3K27me3 in miR-155-overexpressing cells suggesting that miR-155 may restrain terminal differentiation by silencing these TFs via PRC2 (Fig. 1h and Supplementary Fig. 5b). Indeed, with the exception of *Prdm1* and *Id2*, these pro-differentiation TFs were less expressed in KLRG1$^-$ CD62L$^-$ CD8$^+$ T effector cells isolated 5 days after gp100-VV infection (Fig. 1h). Taken together, these findings indicate that miR-155

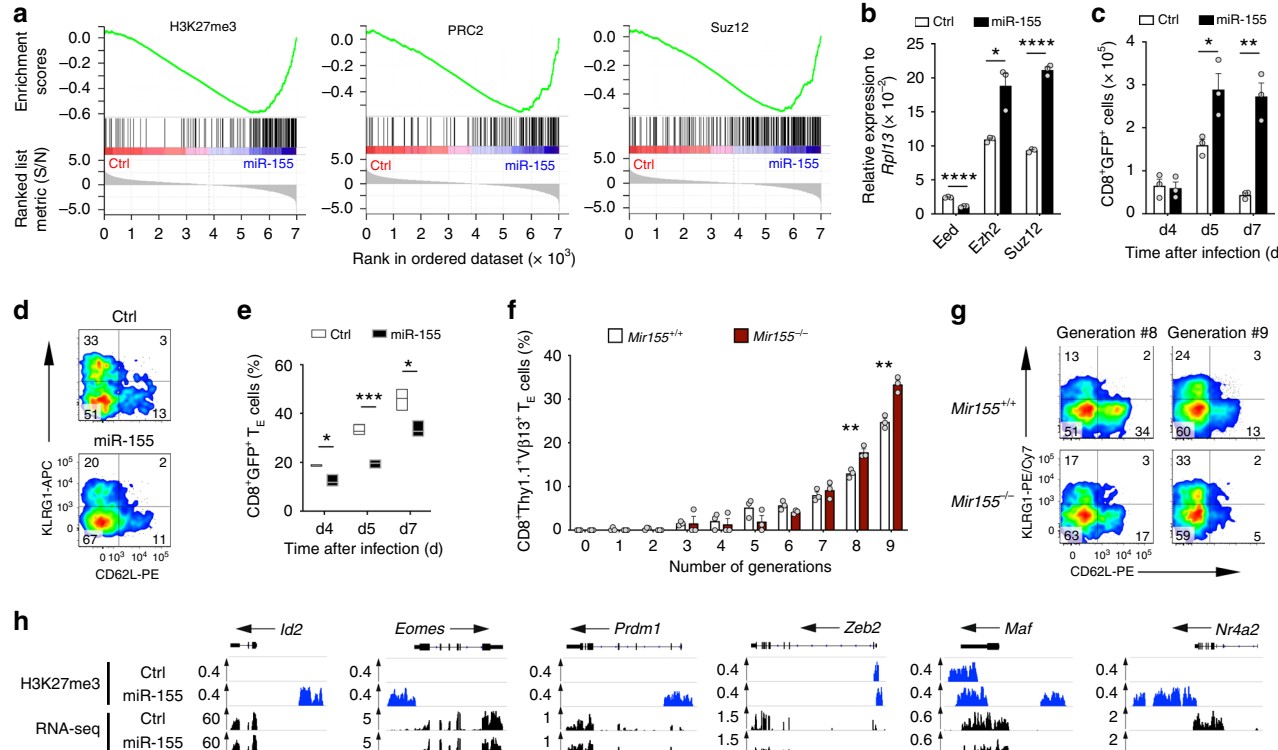

**Fig. 1** miR-155 epigenetically silences CD8+ T cell differentiation. **a** Negative enrichment of H3K27me3 genes[20] (left) and PRC2 (middle) and Suz12 (right) targets[21] in miR-155-overexpressing cells. **b** Quantitative RT-PCR of *Eed*, *Ezh2*, and *Suz12* mRNA in miR-155 and Ctrl-overexpressing cells sorted 5 days following adoptive transfer of $3 \times 10^5$ pmel-1 CD8+ T cells transduced with miR-155 or Ctrl-miR into wild-type mice in conjunction with gp100-VV. Bars (mean ± s.e.m. of technical triplicates) are relative to *Rpl13* mRNA. **c** Number of splenic pmel-1 CD8+GFP+ T cells assessed at different time points after transfer as in **b**. **d** Flow cytometry of splenic pmel-1 CD8+GFP+ T cells 5 days after transfer as in **b**. Numbers indicate the percentage of cells after gating on live CD8+GFP+ T cells. **e** Percentage of terminal effector (KLRG1+CD62L−, $T_E$) in the spleen assessed at different time points after transfer as in **b**. Data are presented as box plots extending to minimum and maximum values. Bands inside the boxes represent median values of three individual mice. **f** Percentage of pmel-1 CD8+Thy1.1+Vβ13+ $T_E$ cells per generation after adoptive transfer of $1.5 \times 10^5$ pmel-1 TCR transduced CFSE-labeled *Mir155*+/+ or *Mir155*−/− CD8+ T cells into Ly5.1 mice in conjunction with gp100-VV. **g** Flow cytometry of CD8+Thy1.1+Vβ13+ T cells in generation #8 and #9 after transfer as in **f**. Numbers indicate the percentage of cells after gating on live CD8+Thy1.1+Vβ13+ T cells. **h** H3K27me3 marks and RNA-seq reads at gene loci known to promote CD8+ T cell terminal differentiation in miR-155 and Ctrl-miR-overexpressing cells. ChIP-seq was performed on KLRG1− pmel-1 CD8+ T cells transduced with miR-155 or Ctrl-miR and cultured in vitro for 5 days. Gene expression was evaluated by RNA-seq of KLRG1−CD62L− cells sorted from transferred cells 5 days after adoptive transfer of $3 \times 10^5$ miR-155 or Ctrl-miR-overexpressing cells into wild-type mice in conjunction with gp100-VV. RNA-seq data were obtained from triplicated groups of three individual mice. Data are representative of at least two independent experiments (**b**−**g**), *P < 0.05; **P < 0.01; ****P < 0.001 (unpaired two-tailed Student's t-test)

inhibits CD8+ T cell terminal differentiation via PRC2-mediated epigenetic mechanisms.

**Ezh2 is required for miR-155-driven T cell antitumor immunity.** The PRC2 subunit Ezh2 is the major histone-lysine N-methyltransferase mediating H3K27 trimethylation[38]. To determine whether the activity of miR-155 requires PRC2 function we overexpressed miR-155 or Ctrl-miR in *Ezh2* sufficient and deficient pmel-1 CD8+ T cells. We then evaluated T cell engraftment, differentiation, cytokine production, and antitumor function after adoptive transfer into B16 tumor-bearing mice in conjunction with gp100-VV administration. As shown above (Fig. 1c), miR-155-overexpressing cells accumulated more robustly than controls (Fig. 2a). However, in the absence of *Ezh2* the accumulation of miR-155-overexpressing cells was dramatically reduced and now comparable to wild-type cells expressing Ctrl-miR (Fig. 2a). Furthermore, the previously seen limiting of terminal differentiation in miR-155 cells was impaired in *Ezh2*−/− cells as the formation of $T_E$ cells increased to control levels (Fig. 2b, c and Supplementary Fig. 6). Ezh2 was also essential for long-term cytokine production and polyfunctionality in miR-155-overexpressing cells. Indeed, deletion of *Ezh2* in

miR-155-overexpressing cells resulted in the rapid loss of effector functions 10 days after transfer (Fig. 2d). Consistent with these findings, miR-155-overexpressing *Ezh2*−/− cells were less effective at controlling tumor growth and prolonging mice survival than miR-155-overexpressing wild-type controls (Fig. 2e, f). Taken together, these data indicate that the PRC2 catalytic unit, Ezh2, is critical for miR-155 effector function.

**Jarid2 deletion does not mimic miR-155 functional benefits.** We next sought to determine how miR-155 promoted PRC2 and Ezh2 activity. Jumonji, AT rich interactive domain 2 (Jarid2) is the most well-characterized PRC2 cofactor and an established target of miR-155[39] (Fig. 3a). Jarid2 critically recruits PRC2 to specific chromatin loci and has been reported to have both stimulatory and inhibitory functions on PRC2 catalytic activity depending on chromatin context[38], but its role in CD8+ T cells is unknown. We hypothesized that in CD8+ T cells, miR-155 downregulates Jarid2 expression, releasing Jarid2-mediated PRC2 inhibition. Consistently, we detected decreased Jarid2 protein in miR-155-overexpressing cells (Fig. 3b). To test whether deletion of *Jarid2* provides the functional advantages conferred by miR-155 overexpression, we deleted *Jarid2* in pmel-1 *Jarid2*fl/fl

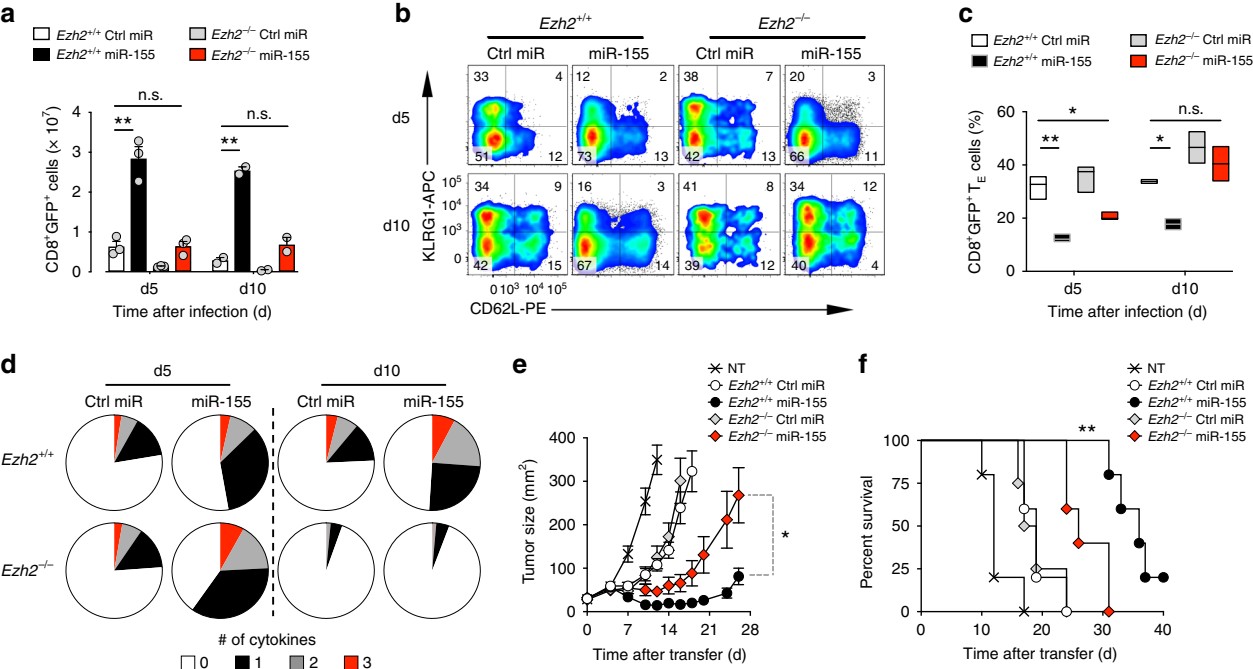

**Fig. 2** Ezh2 is required for the enhanced CD8+ T cell antitumor immunity conferred by miR-155. **a** Number of live CD8+GFP+ T cells in the spleen assessed at day 5 and 10 following adoptive transfer of $3 \times 10^5$ pmel-1 $Ezh2^{+/+}$ or $Ezh2^{-/-}$ CD8+ T cells transduced with miR-155 or Ctrl-miR into B16 tumor-bearing mice in conjunction with gp100-VV. Bars represent the mean ± s.e.m. of three or two individual mice. **b** Flow cytometry of splenic CD8+GFP+ T cells 5 and 10 days after transfer as in **a**. Numbers indicate the percentage of cells after gating on live CD8+GFP+ T cells. **c** Percentage of splenic pmel-1 CD8+GFP+ $T_E$ cells at indicated time points after transfer. Data are presented as box plots extending to minimum and maximum values. Bands inside the boxes represent median values of three individual mice. **d** Combinatorial cytokine response of splenic CD8+GFP+ T cells 5 and 10 days after transfer as in **a**, as determined by the Boolean combination of gates identifying IFN-γ+, IL-2+, and TNF+ cells. Data are presented as the mean of three individual mice. **e**–**f** Tumor size (mean ± s.e.m.) (**e**) and survival curve (**f**) of B16 tumor-bearing mice after adoptive transfer of $4 \times 10^6$ cells generated as in **a** in conjunction with gp100-VV ($n = 5$ mice/group). NT no treatment. Data are representative of two independent experiments. *$P < 0.05$; **$P < 0.01$ (unpaired two-tailed Student's $t$-test) (**a**, **d**), **$P < 0.01$ [a Log-rank (Mantel-Cox) Test] (**f**)

CD8+ T cells with a GFP-Cre retroviral vector (Fig. 3c). We then transferred the cells into wild-type mice infected with gp100-VV and evaluated T cell expansion and differentiation. Surprisingly, we found no significant differences in cell accumulation between $Jarid2^{-/-}$ and wild-type controls (Fig. 3d, e). Likewise, deletion of $Jarid2$ had no major influence on CD8+ T cell differentiation and effector functions (Fig. 3f, g). If anything, $Jarid2$-deficient cells tended to undergo terminal differentiation faster than controls, based on the increased frequency of $T_E$ cells on day 5 post-infection (Fig. 3f, g). These findings indicate that Jarid2 plays a marginal role in CD8+ T cell responses to viral infection. To further evaluate the contribution of Jarid2 in CD8+ T cell antitumor immunity we transferred pmel-1 $Jarid2^{-/-}$ CD8+ T cells into tumor-bearing mice in conjunction with gp100-VV and interleukin-2 (IL-2) administration. Also in this setting, $Jarid2$ deletion did not provide functional advantages to T cells (Fig. 3h). Instead, $Jarid2^{-/-}$ cells were less effective than their wild-type counterpart at controlling tumor growth and extending mice survival (Fig. 3h). Collectively, these results indicate that $Jarid2$ deletion does not phenocopy miR-155 overexpression but rather has subtle antithetical effects on T cell differentiation and function. Thus, the enhanced PRC2 activity observed in miR-155-overexpressing cells must have relied on an alternative molecular mechanism to Jarid2 downregulation.

**Phf19 is a key downstream factor of miR-155 in CD8+ T cells.** To identify potential downstream targets of miR-155 involved in PRC2 function, we re-examined our RNA-seq data set

comparing KLRG1−CD62L− cells overexpressing miR-155 and Ctrl-miR. We noticed that $Phf19$—a recently discovered Polycomb-like protein that recruits PRC2 to genomic targets by binding to histone methyl-lysine[40,41]—was strongly upregulated in miR-155-overexpressing cells (Fig. 4a and Supplementary Fig. 1b). This observation was further validated by qPCR analysis, demonstrating that miR-155-overexpressing cells contained nearly 3-fold higher $Phf19$ transcripts than controls (Fig. 4b). Conversely, $Phf19$ levels were significantly reduced in miR-155-deficient CD8+ T cells compared to wild-type cells (Fig. 4c). $Phf19$ has been reported to be upregulated by pAkt in cancer cells[42] and we and others have previously showed that Akt signaling is heightened by miR-155 through repression of inositol polyphosphate-5-phosphatase D (also known as Ship1)[17,18] (Fig. 4d, e and Supplementary Fig. 7). To determine whether miR-155 induced $Phf19$ by enhancing Akt signaling, we first measured $Phf19$ expression in miR-155 and Ctrl-miR-overexpressing cells after incubation with the AKT inhibitor VIII or transduction with a constitutively active form of Akt (AktCA)[18]. Consistent with our prior findings, miR-155 upregulated $Phf19$ (Fig. 4f, g). Strikingly, blockade of AKT signaling significantly inhibited $Phf19$ expression (Fig. 4f) whereas constitutive Akt signaling drove $Phf19$ expression to saturation, abrogating any further upregulation of $Phf19$ by miR-155 (Fig. 4g). Next, we tested whether downregulation of Ship1 alone would induce $Phf19$. To delete Ship1 in CD8+ T cells, we transduced Cas9+ CD8+ T cells with a retroviral vector encoding a sgRNA targeting Ship1 exon 5 (Fig. 4h). Ship1 knockdown resulted in both enhanced pAkt levels and a substantial

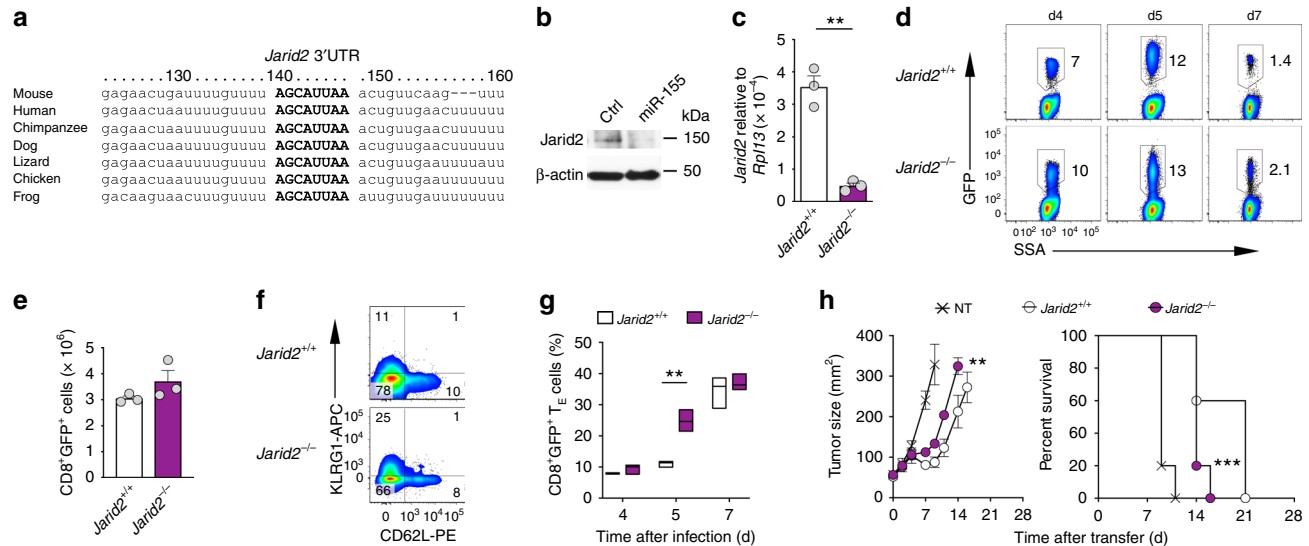

**Fig. 3** Deletion of Jarid2 does not recapitulate the functional advantages conferred by miR-155 in CD8+ T cells. **a** Sequence alignment of the *Jarid2* 3′ UTR in multiple species. The predicted miR-155 target site sequence is shown in capital bold. **b** Jarid2 protein level in miR-155-overexpressing cells assessed by Immunoblot. **c** Quantitative RT-PCR of *Jarid2* mRNA in pmel-1 *Jarid2*+/+ and pmel-1 *Jarid2*−/− cells. Bars (mean ± s.e.m. of technical triplicates) are relative to *Rpl13* mRNA. **d** Flow cytometry of splenic CD8+GFP+ T cells assessed at different time points following adoptive transfer of $3 \times 10^5$ pmel-1 *Jarid2*fl/fl CD8+ T cells transduced with GFP-Cre or Ctrl GFP into wild-type mice in conjunction with gp100-VV. Numbers indicate the percentage of cells after gating on live CD8+GFP+ T cells. **e** Number of splenic pmel-1 CD8+GFP+ T cells at day 5 after transfer as in **d**. Bars represent the mean ± s.e.m. of three individual mice. **f** Flow cytometry of splenic pmel-1 CD8+GFP+ T cells 5 days after transfer as described in **d**. Numbers indicate the percentage of cells after gating on live CD8+GFP+ T cells. **g** Percentage of splenic pmel-1 CD8+GFP+ $T_E$ cells at indicated time points after transfer as in **d**. Data are presented as box plots extending to minimum and maximum values. Bands inside the boxes represent median values of three individual mice. **h** Tumor size (left, mean ± s.e.m.) and survival curve (right) of B16 tumor-bearing mice after adoptive transfer of $2 \times 10^6$ cells in conjunction with gp100-VV and IL-2 ($n = 5$ mice/group). NT no treatment. Data are representative of two independent experiments. **$P < 0.01$ (unpaired two-tailed Student's t-test) (**c**, **g**, and **h**: left); ***$P < 0.005$ [a Log-rank (Mantel-Cox) Test] (**h**: right)

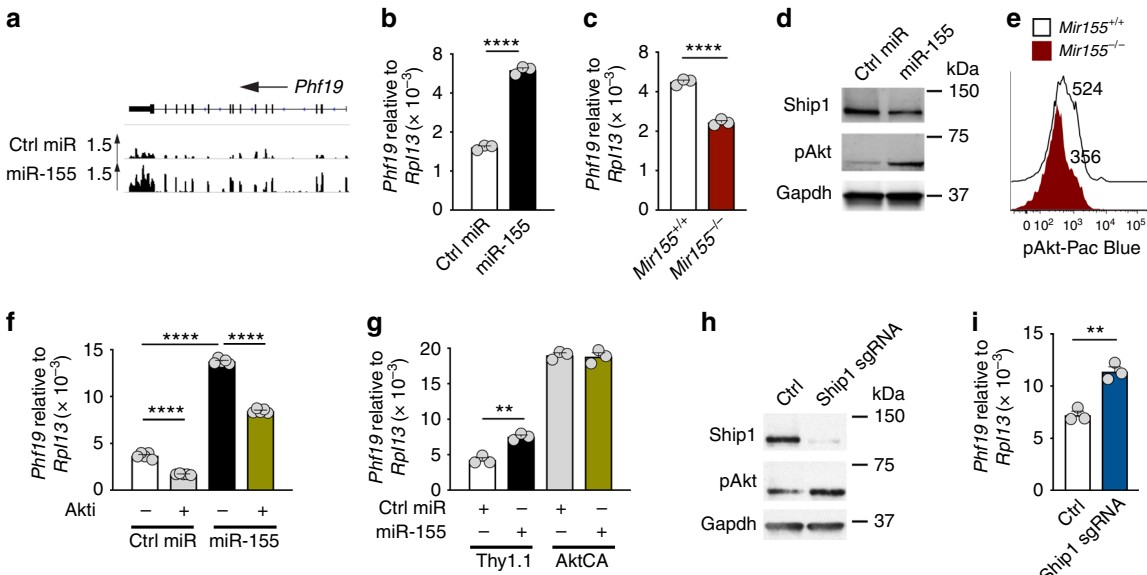

**Fig. 4** miR-155 promotes *Phf19* expression by enhancing Akt signaling via downregulation of Ship1. **a**, **b** RNA-seq reads (**a**) and quantitative RT-PCR (**b**) of *Phf19* mRNA in miR-155 and Ctrl-miR-overexpressing cells. Bars (mean ± s.e.m. of technical triplicates) are relative to *Rpl13* mRNA. **c** Quantitative RT-PCR of *Phf19* mRNA in in vitro activated KLRG1− miR-155 sufficient and deficient CD8+ T cells. Bars (mean ± s.e.m. of technical triplicates) are relative to *Rpl13* mRNA. **d** Ship1 and pAkt levels in miR-155-overexpressing cells assessed by Immunoblot. **e** pAkt levels of in vitro activated KLRG1− miR-155 sufficient and deficient CD8+ T cells assessed by flow cytometry. **f**, **g** Quantitative RT-PCR of *Phf19* mRNA in ex vivo sorted CD8+ T cells overexpressing miR-155 and Ctrl-miR after a 6 h incubation with or without AKT inhibitor VIII (Akti) (**f**) or transduction with constitutively active Akt (AktCA) or Thy1.1 control (**g**). Bars represent the mean ± s.e.m. of technical triplicates. **h** Ship1 and pAkt levels in Cas9+ CD8+ T cells transduced with Ship1-specific gRNA assessed by Immunoblot. **i** Quantitative RT-PCR of *Phf19* mRNA in Cas9+ CD8+ T cells transduced with Ship1-specific gRNA or control. Bars represent the mean ± s.e.m. of technical triplicates. Data are representative of two independent experiments. *$P < 0.05$; **$P < 0.01$; ****$P < 0.001$ (unpaired two-tailed Student's t-test)

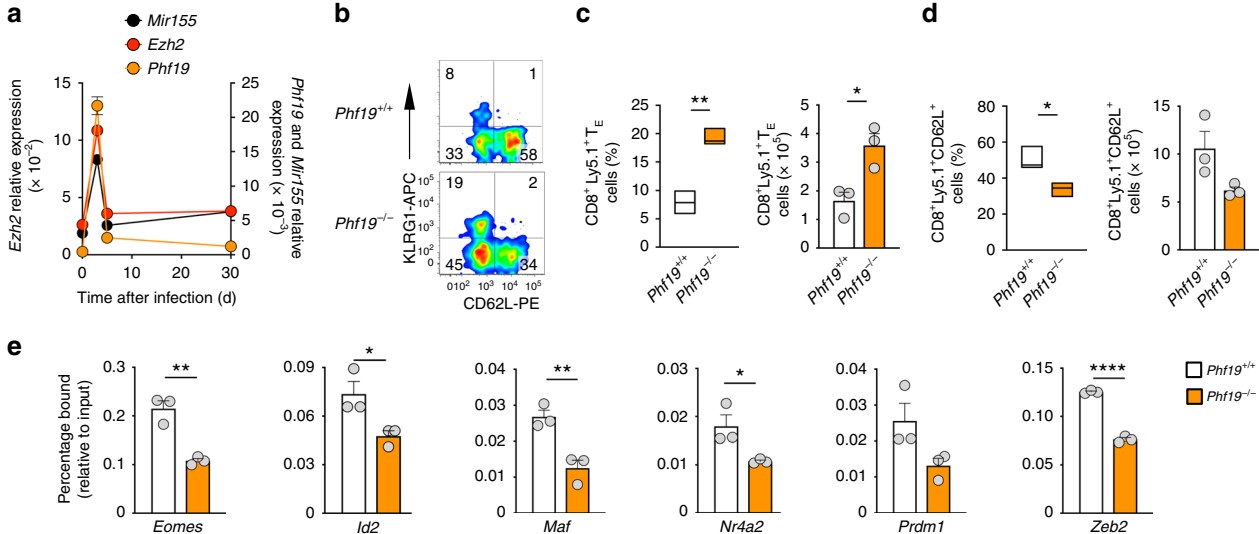

**Fig. 5** Phf19 restricts CD8$^+$ T cell terminal differentiation. **a** Quantitative RT-PCR of *Mir155*, *Ezh2*, and *Phf19* after transfer of 10$^5$ pmel-1 CD8$^+$Ly5.1$^+$ T cells into wild-type mice in conjunction with gp100-VV assessed at the indicated points. *Ezh2* and *Phf19* levels are relative to *Rpl13*, *Mir155* levels are relative to *U6*. **b** Flow cytometry of splenic live CD8$^+$Ly5.1$^+$ T cells 5 days after transfer of 3 × 10$^5$ naïve pmel-1 CD8$^+$Ly5.1$^+$ *Phf19$^{+/+}$* or *Phf19$^{-/-}$* T cells into wild-type mice in conjunction with gp100-VV. Numbers indicate percentage after gating on live CD8$^+$Ly5.1$^+$ T cells. **c** Percentage (left) and number (right) of splenic pmel-1 CD8$^+$Ly5.1$^+$ T$_E$ cells 5 days after transfer as in **b**. **d** Percentage (left) and number (right) of splenic CD8$^+$Ly5.1$^+$CD62L$^+$ T cells 5 days after adoptive transfer as in **b**. Data are presented as box plots extending to minimum and maximum values. Bands inside the boxes represent median values of three individual mice. **e** ChIP-qPCR using H3K27me3 antibody on in vitro activated non T$_E$ KLRG1$^-$ *Phf19$^{+/+}$* or *Phf19$^{-/-}$* T cells with primers specific to the transcription start site of selected TFs. ChIP enrichments are presented as the percentage of protein bound, normalized to input. Bars represent the mean ± s.e.m. of technical triplicates. Data are representative of two independent experiments. *$P < 0.05$; **$P < 0.01$; ****$P < 0.001$ (unpaired two-tailed Student's *t*-test)

upregulation of *Phf19* expression in CD8$^+$ T cells (Fig. 4h, i). Altogether, these results indicate that miR-155 overexpression induced *Phf19* transcription indirectly by promoting Akt signaling through downregulation of Ship1.

The role of Phf19 in immunity and particularly in CD8$^+$ T cell biology is unknown. We sought first to determine if Phf19 expression was dynamically regulated in CD8$^+$ T cells responding to gp100-VV infection. We found that *Phf19* was strongly induced at the early stages of acute immune response, sharply downregulated at peak effector response and maintained at low levels throughout transition to memory phase (Fig. 5a). These findings suggested a potential role of Phf19 in regulating CD8$^+$ T cell effector differentiation. The rapid spike of induction was similarly observed for *Ezh2*, indicating coordinated expression of PRC2 and its associated factor Phf19 during the immune response (Fig. 5a). Remarkably, *Mir155* followed a virtually identical pattern of expression, emphasizing the interplay between these molecules during physiologic immune response (Fig. 5a). To investigate whether Phf19 phenocopies the effects of miR-155 in restraining CD8$^+$ T cell differentiation, we evaluated the induction of T$_E$ cells in *Phf19$^{-/-}$* cells after transfer of naive CD8$^+$ T cells into wild-type mice infected with gp100-VV. Although *Phf19$^{-/-}$* T cells were derived from germline knockouts, we did not find gross alterations in T cell development and peripheral homeostasis in *Phf19*-deficient animals (Supplementary Fig. 8). We found that *Phf19*-deficient T cells were prone to undergo terminal differentiation as shown by both the increased frequency and number of T$_E$ cells (Fig. 5b, c) and the reduced percentage and number of KLRG1$^-$CD62L$^+$ memory precursors (Fig. 5b, d). Consistent with this defect in memory precursor formation, we observed reduced frequency and absolute number of memory cells in *Phf19$^{-/-}$* cells, although no major difference in the distribution of memory subsets was detected (Supplementary Fig. 9).

To test whether Phf19 promoted the silencing of the pro-effector and pro-exhaustion TFs that were suppressed by miR-155 overexpression (Fig. 1h and Supplementary Fig. 5), we performed a H3K27me3 ChIP-qPCR analysis on KLRG1$^-$ T cells in the presence and absence of *Phf19*. We observed in *Phf19$^{-/-}$* CD8$^+$ T cells a reduced deposition of H3K27me3 at all previously shown TFs targeted in miR-155-overexpressing cells (Fig. 5e), suggesting that Phf19 and miR-155 regulate a common core molecular program. Next, to determine the extent of the overlap between the transcriptional profiles of Phf19 and miR-155, we compared the transcriptome of KLRG1$^-$CD62L$^-$ cells from *Phf19$^{-/-}$* and miR-155-overexpressing cells isolated at peak immune response. There were 346 genes significantly changed in *Phf19$^{-/-}$* compared to wild-type cells. Strikingly, 166 of these genes were also differentially regulated in miR-155-overexpressing cells. Specifically, nearly 65% of the genes downregulated in *Phf19$^{-/-}$* cells were upregulated in miR-155-overexpressing cells ($P = 2.2e-16$) (Fig. 6a, b). Conversely, about 1/6 of the genes upregulated in *Phf19$^{-/-}$* cells were downregulated in miR-155-overexpressing cells ($P = 4.9e-11$) (Fig. 6a, b). The broad overlap of the Phf19 transcriptional program with that of miR-155 was further supported by GSEA, which revealed that numerous gene sets displayed opposite enrichment pattern in miR-155-overexpressing and *Phf19*-deficient cells (Fig. 6c, d and Supplementary Data 4, 5). Among 32 datasets positively enriched (FDR q < 0.25) in *Phf19$^{-/-}$* cells, >50% were also negatively enriched in miR-155-overexpressing cells ($P = 1.43e-6$). Remarkably, 1012 of 1113 datasets negatively enriched (FDR q < 0.25) in *Phf19$^{-/-}$* cells were positively enriched in miR-155-overexpressing cells ($P = 2.2e-16$) (Fig. 6c). For instance, genes upregulated in primary versus secondary CD8$^+$ T cell responses against LCMV infection[43], which represent a molecular signature of less-differentiated cells[44], were enriched in miR-155-overexpressing cells but depleted in

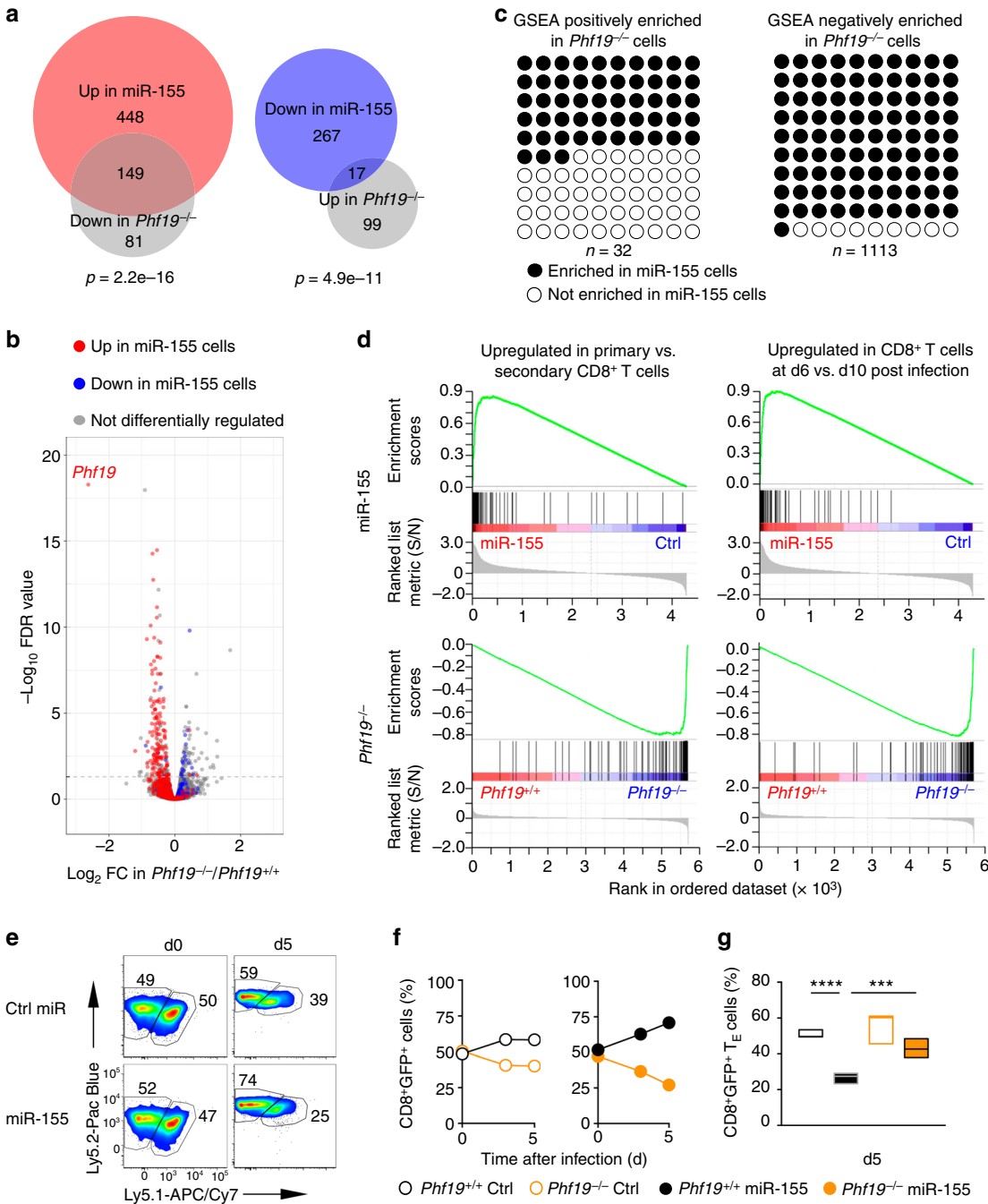

**Fig. 6** Phf19 is a critical downstream factor of miR-155 in CD8+ T cells. **a** Venn diagrams and **b** volcano plot depicting the number of overlapping differentially expressed genes between differentially expressed genes in miR-155-overexpressing and $Phf19^{-/-}$ KLRG1⁻CD62L⁻ CD8+ T cells. Gene expression was evaluated by RNA-seq of pmel-1 KLRG1⁻CD62L⁻ CD8+ T cells 5 days after adoptive transfer of $3 \times 10^5$ pmel-1 $Phf19^{+/+}/Phf19^{-/-}$ cells or pmel-1 cells overexpressing Ctrl-miR/miR-155 into wild-type mice in conjunction with gp100-VV. RNA-seq data were obtained from triplicated groups of three individual mice. **c** Gene sets significantly enriched (FDR < 0.25) in $Phf19^{-/-}$ CD8+ T cells. Gene sets also enriched in miR-155 overexpressing cells are highlighted in black. **d** Enrichment of genes upregulated in CD8+ T cells responding to primary *vs* secondary LCMV infections[43] (left panels) and enrichment of genes upregulated in CD8+ T cells at d6 vs d10 post LmOVA infections[45] (right panels) in miR-155-overexpressing and $Phf19^{-/-}$ CD8+ T cells. **e** Flow cytometry analysis of congenically-distinguishable live pmel-1 $Phf19^{+/+}$ Ly5.2+/+ and pmel-1 $Phf19^{-/-}$ Ly5.1+/− cells transduced with either miR-155 or Ctrl-miR assessed pre-transfer and 5 days after co-transfer of $3 \times 10^5$ cells into wild-type mice in conjunction with gp100-VV administration. Numbers adjacent to outlined areas indicate percentage after gating on live CD8+GFP+ T cells. **f, g** Percentages of live pmel-1 CD8+GFP+ T cells transduced with Ctrl-miR (left panel) or miR-155 (right panel) (**f**) and CD8+ GFP+ TE cells (**g**) at indicated time points after transfer as described in **e**. Symbols represent the mean ± s.e.m. of three individual mice; small horizontal lines (right panel) indicate the mean ± s.e.m. Data are representative of two independent experiments. ***P < 0.005; ****P < 0.001 (unpaired two-tailed Student's t-test)

*Phf19*⁻/⁻ cells (Fig. 6d, left panels). Likewise, genes upregulated in d6 versus d10 post LmOVA infections[45], which represent a molecular signature of more-proliferative cells, were enriched in miR-155-overexpressing cells but depleted in *Phf19*⁻/⁻ cells (Fig. 6d, right panels). Taken together, these findings indicate that Phf19 orchestrates a transcriptional program extensively shared with miR-155, establishing Phf19 as a critical downstream factor of miR-155.

Finally, to test whether the enhanced immune response mediated by miR-155 overexpression were dependent on Phf19 function, we co-transferred *Phf19* sufficient and deficient cells that were transduced with either miR-155 or Ctrl-miR into wild-type recipients infected with gp100-VV. While the lack of *Phf19* had measurable impact on both expansion and differentiation of Ctrl-miR cells, these differences where considerably heightened in cells overexpressing miR-155 (Fig. 6e–g). Taken together, our data indicate Phf19 is a critical downstream factor of miR-155 in mediating T cell expansion and restricting senescence.

**Phf19 epigenetically reprograms T cell antitumor immunity**. We next sought to evaluate whether Phf19 overexpression could be used to epigenetically reprogram T cell function and

differentiation in a more specific and effective fashion than miR-155. Because a chromatin-independent role of Polycomb-like proteins has been reported[46], we generated a Phf19 mutant (Phf19mut), with attenuated chromatin-binding capacity to test whether the effects mediated by Phf19 overexpression were dependent on epigenetic mechanisms. To this end, we mutated conserved tryptophan (W41) and tyrosine (Y47) in the aromatic cage of the Phf19 Tudor domain, which has previously been shown to mediate human PHF19 chromatin binding[40,41,46,47] (Fig. 7a). Consistently, we detected considerably less Phf19 protein in the chromatin fraction when CD8⁺ T cells were transduced with Phf19mut (Fig. 7b). As reported in other cell types[47], overexpression of Phf19 but not Phf19mut enhanced Ezh2 association to chromatin and H3K27me3 deposition in CD8⁺ T cells (Supplementary Fig. 10). We then adoptively transferred pmel-1 cells transduced with Phf19, Phf19mut, or Thy1.1 into wild-type mice and measured T cell expansion, function, and differentiation in response to gp100-VV infection. Reminiscent of miR-155-overexpressing cells, Phf19-transduced cells expanded more robustly than Thy.1.1 controls (Fig. 7c, d) and displayed limited senescence (Fig. 7e, f) and sustained cytokine production (Fig. 7g, h). These functional advantages were almost abrogated in cells transduced with Phf19mut (Fig. 7c–h), indicating that

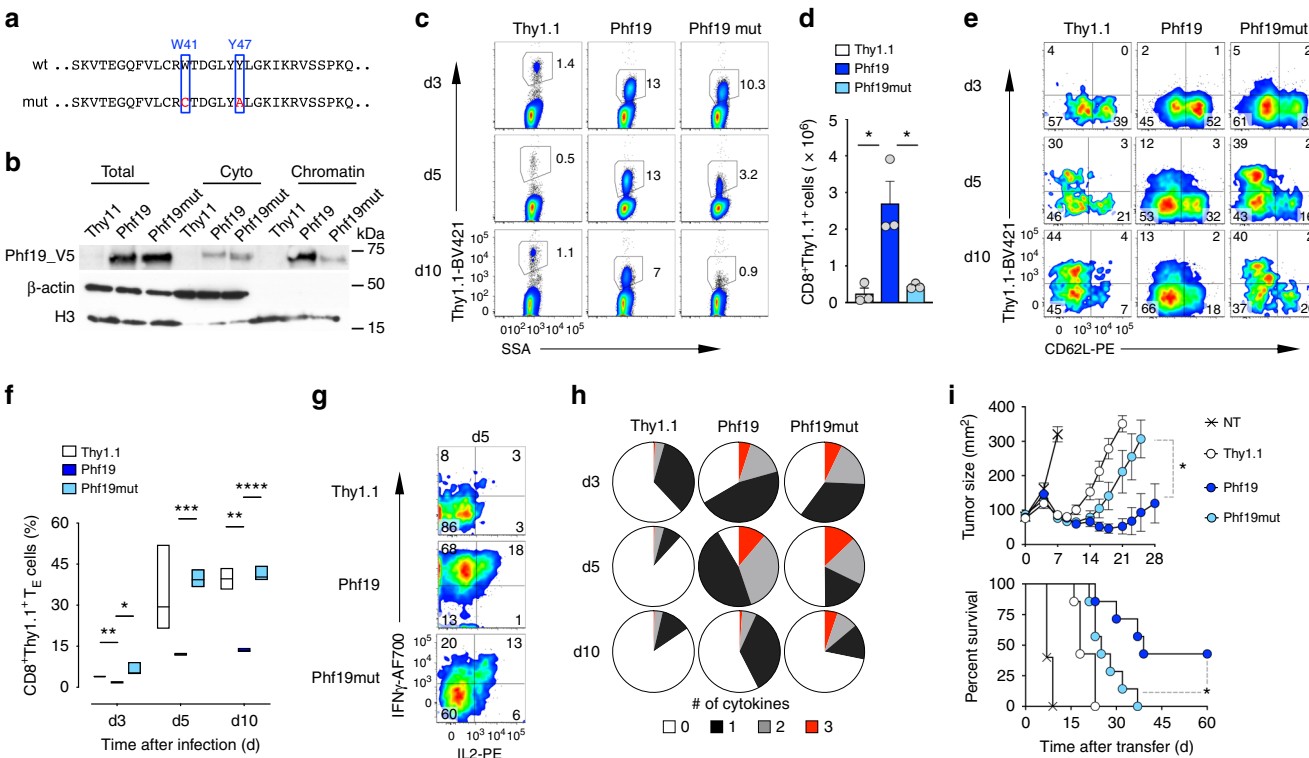

**Fig. 7** Phf19 epigenetically reprograms T cell antitumor immunity. **a** Amino acid sequence alignment of the partial aromatic cage of Phf19. Tryptophan (W) and tyrosine (Y) residues were mutated to cysteine (C) or alanine (A), respectively. **b** Immunoblot of total, soluble, and chromatin-bound proteins from CD8⁺ T cells transduced with the Phf19Thy1.1, Phf19mutThy1.1, and Thy1.1 control. **c** Flow cytometry of splenic T cells at indicated time points following adoptive transfer of 3 × 10⁵ pmel-1 CD8⁺ T cells transduced with Phf19Thy1.1, Phf19mutThy1.1, or Thy1.1 into wild-type mice in conjunction with gp100-VV. Numbers indicate percentage after gating on live CD8⁺ Thy1.1⁺ T cells. **d** Number of splenic pmel-1 CD8⁺Thy1.1⁺ T cells 5 days after transfer as in **c**. Bars represent the mean ± s.e.m. of three individual mice. **e** Flow cytometry of splenic CD8⁺Thy1.1⁺ T cells at indicated time points after transfer as in **c**. Numbers indicate percentage after gating on live CD8⁺Thy1.1⁺ T cells. **f** Percentage of splenic pmel-1 CD8⁺Thy1.1⁺ T_E cells at indicated time points as in **c**. Data are presented as box plots extending to minimum and maximum values. Bands inside the boxes represent median values of three individual mice. **g** Intracellular cytokine staining of splenic CD8⁺Thy1.1⁺ T cells 5 days after transfer as in **c**. Numbers indicate percentage after gating on live CD8⁺Thy1.1⁺ T cells. **h** Combinatorial cytokine response of splenic CD8⁺Thy1.1⁺ T cells as in **c**, as determined by the Boolean combination of gates identifying IFN-γ⁺, IL-2⁺, and TNF⁺ cells. Data are presented as the mean of three individual mice. **i** Tumor size (left, mean ± s.e.m.) and survival curve (right) of B16 tumor-bearing mice receiving 2 × 10⁶ cells generated as in **c** in conjunction with gp100-VV and IL-2 (*n* = 7 mice/group). NT no treatment. Data are representative of two independent experiments. *$P < 0.05$; **$P < 0.01$; ***$P < 0.005$; ****$P < 0.001$ (unpaired two-tailed Student's *t*-test) (**d**, **h**, and **i** left); *$P < 0.05$ (a Log-rank (Mantel-Cox) Test) (**i** right)

Phf19 activity was dependent on its capacity to remodeling chromatin. Lastly, to evaluate whether the epigenetic reprogramming mediated by Phf19 result in augmented antitumor function we adoptively transferred pmel-1 cells overexpressing Phf19, Phf19mut, or Thy1.1 into tumor-bearing mice in conjunction with administration of gp100-VV and IL-2. We found that Phf19 cells mediated a dramatic and long-lasting antitumor response resulting in increased mice survival compared to Thy1.1 controls (Fig. 7i). These therapeutic benefits, however, were greatly reduced in mice receiving Phf19mut cells (Fig. 7i), indicating that the chromatin-binding capacity of Phf19 was critical for the enhanced CD8+ T cell antitumor immunity. This conclusion was further strengthened by the finding that the therapeutic impact of Phf19 overexpression was abolished in CD8+ T cells lacking *Ezh2* (Supplementary Fig. 11). Altogether these findings demonstrate that Phf19 potentiates CD8+ T cell antitumor activity by epigenetic reprogramming via PRC2.

## Discussion

In this study, we demonstrate that miR-155 enhances the antitumor response by epigenetically restricting CD8+ T cell differentiation and functional exhaustion. We show that miR-155 promotes PRC2 activity to silence key TFs known to drive terminal differentiation and exhaustion. While the miR-155–Jarid2–PRC2 axis has been shown to play a pivotal role in Th17 cell function[39], this axis has a minor inhibitory role in CD8+ T cell immune responses to both virus and cancer. Instead, we demonstrate that miR-155 influences PRC2 function by indirectly inducing Phf19 expression (Fig. 8). These findings reveal an unprecedented complexity of the miRNA–epigenetic circuitry. Not only do miRNAs modulate the epigenetic landscape by directly targeting key components of the epigenetic machinery[10], but also affect key signaling pathways to regulate their expression. Consistent with its role in ESC[40], we identify Phf19 as a gatekeeper of CD8+ T cell differentiation: CD8+ T cells

deficient of *Phf19* undergo terminal differentiation at an accelerated rate in response to viral infection, whereas CD8+ T cells overexpressing this molecule are preferentially maintained in a less-differentiated stem cell−like memory state[48]. Although Polycomb-like protein can exhibit chromatin-independent activities[46], the role of Phf19 in restricting CD8+ T cell differentiation is largely dependent on its chromatin-binding capacity and regulation of pro-differentiating TFs via PRC2. Interestingly, these TFs including *Eomes*, *Id2*, *Prdm1*, *Zeb2*, *Maf*, and *Nr4a2* have also been reported to be Phf19 binding-targets in ESC[40], emphasizing the existence of a conserved molecular network regulating self-renewal and differentiation in stem cells and T lymphocytes[2].

The role of Ezh2 and PRC2 in CD8+ T cell differentiation has just begun to be addressed. While Kakaradov et al.[49] did not observe significant differences in the formation of KLRG1+ effectors in the absence of *Ezh2* at the time point analyzed, Gray et al.[50] reported that Ezh2 drives CD8+ T cell terminal differentiation and guides the loss of memory T cell potential and multipotency. Their findings are in striking contrast to our observations and a recent report by He et al.[51] indicating that Ezh2 is required to establish memory CD8+ T cells by silencing pro-differentiation TFs—a conclusion in line with the role of PRC2 in stem cells where it inhibits lineage-determining genes[22]. These inconsistencies are likely due to key differences in experimental setting. For instance, we employed adoptive transfer experiments to specifically evaluate the role of PRC2 in CD8+ T cells, whereas Gray et al.[50] studied LCMV-specific CD8+ T cell responses in hosts deficient of *Ezh2* in both CD8+ and CD4+ T cell compartments, making the cell intrinsic role of *Ezh2* in CD8+ T cells unclear. Moreover, while in our study *Ezh2* is deleted in naive CD8+ T cells prior to any antigenic stimulation, Gray et al.[50] mostly employed a *Gzmb*-Cre system, which won't delete *Ezh2* until granzyme B is expressed in effector cells. Since memory and effector cell fates are established at early times post-infection[52] and possibly even within the first cell division[53], the delayed deletion of *Ezh2* in *Gzmb*+ cells may have led to neglecting its contribution to restraining terminal differentiation. Indeed, H3K27me3 levels are elevated in naïve and central memory CD8+ T cells relative to effector memory CD8+ T cells[54], indicating that PRC2 activity is predominant in less-differentiated T cells. Specifically, in naive CD8+ T cells, H3K27me3 marks heavily decorate genes promoting effector development and function[54–56], suggesting that PRC2 is required to limit T cell differentiation. These repressive marks, however, are quickly lost upon T cell activation, well before any evidence of *Gzmb* expression[55]. Recent findings showing discordant apoptosis of *Cd4*Cre*Ezh2*fl/fl and *Gzmb*Cre*Ezh2*fl/fl CD8+ T cells after activation reinforce the critical impact of the deletion time of *Ezh2*[57]

While Phf19 and Ezh2 both play a critical role in limiting CD8+ T cell terminal differentiation downstream of miR-155, we show that Ezh2 has a more pronounced function in controlling CD8+ T cell accumulation and survival. Considering that miR-155 influences the expression of numerous molecules affecting PRC2 activity, it is conceivable that additional players beside Phf19 might also contribute to heightened PRC2 function in miR-155-overexpressing cells. For instance, the transcriptional corepressor LCOR, a recently identified PRC2-associated factor[38] and predicted miR-155 target, is a potential candidate for future investigation due to its documented capacity to drive differentiation of normal and cancerous mammary stem cells[58].

Finally, our findings have considerable implications for the development of new and more effective cancer immunotherapy. T cell antitumor responses are often limited by functional, proliferative, and survival defects associated with terminal

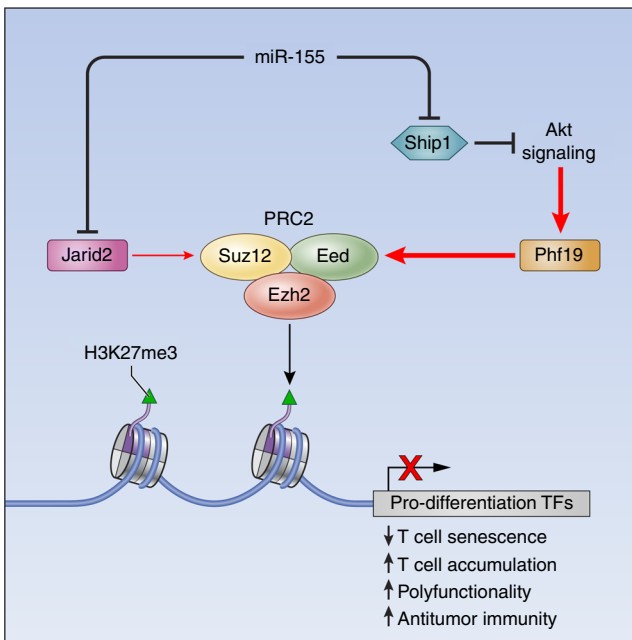

**Fig. 8** The miR-155-PRC2 circuitry potentiates antitumor immunity by epigenetically reprogramming CD8+ T cell fate. Cartoon depicting the regulatory circuitry by which miR-155 epigenetically reprograms CD8+ T cell fate and function via enhancement of PRC2 activity. TFs, Transcription factors

differentiation and exhaustion[1,2]. PRC2 activity in T cells has been shown to be essential for maintaining T cell poly-functionality and promoting persistence, but its function in the tumor microenvironment can be inhibited by Ezh2-specific miRNAs that are elevated under limited availability of glu-cose[14]. Harnessing miR-155 and Phf19 might represent an effective strategy to enhance PRC2 function in tumor-reactive T cells to sustain engraftment, cytokine production, and ulti-mately antitumor immunity. The therapeutic potential of tar-geting PRC2 is further supported by the observation that both overall and disease-free survival in patients with ovarian cancer highly correlate with the number of Ezh2$^+$ CD8$^+$ T cells infil-trating the tumor[14]. Thus, epigenetic reprogramming of CD8$^+$ T cell fate is a promising new avenue for potentiating cancer immunotherapy.

## Methods

**Mice and tumor lines.** C57BL/6 mice were from Charles River; Pmel-1 (B6.Cg-Thy1a/Cy Tg (TcraTcrb)8Rest/J) mice, B6.Cg-Mir155tm1.1Rsky/J (bic/miR-155-) mice, and B6J.129(Cg)-Gt(ROSA)26Sortm1.1(CAG-cas9*,-EGFP) mice were from the Jackson Laboratory; Pmel-1 Ezh2$^{fl/fl}$ Cd4cre mice were provided by Zhang Y. from Temple University. Jarid2$^{fl/fl}$ mice were provided by Muljo S. from NIAID, NIH. Phf19$^{-/-}$ mice were obtained from MBP, UC Davis. Jarid2$^{fl/fl}$, and Phf19$^{-/-}$ mice were crossed with Pmel-1 transgenic animals. B16 (H-2$^b$), a gp100$^+$ mouse melanoma, was from the National Cancer Institute Tumor Repository. B16 (H-2$^b$)-hgp100 was obtained from Hanada K. from NCI, NIH. All mouse experiments were done with the approval of the National Cancer Institute Animal Use and Care Committee.

**Flow cytometry and cell sorting.** All antibodies used for flow cytometry and cell sorting are listed in Supplementary Table 2. Leukocyte Activation Cocktail con-taining phorbol myristate acetate and ionomycin (BD Biosciences) was used for the stimulation of T cells for intracellular cytokine staining. Annexin V staining was performed with Annexin V Apoptosis Detection Kit (eBiosciences). BrdU staining was performed with BrdU Staining Kit (eBiosciences) following the protocol provided by the manufacturer. A LSR II or LSRFortessa (BD Biosciences) was used for flow cytometry acquisition. Samples were acquired with BD FACSDiva (BD Biosciences) and analyzed with FlowJo software (TreeStar). CD8$^+$ GFP$^+$ T cells, CD8$^+$ GFP$^+$KLRG1$^-$CD62L$^-$ T cells, CD8$^+$ GFP$^+$KLRG1$^+$CD62L$^-$ T cells, CD8$^+$ GFP$^+$ Thy1.1$^+$ T cells, CD8$^+$ Ly5.1$^+$KLRG1$^-$CD62L$^-$ T cells, and CD8$^+$ Ly5.1$^+$KLRG1$^+$CD62L$^-$ T cells, were sorted with a FACSAria (BD Bios-ciences). Gating strategies followed for flow cytometry analysis and cell sorting are shown in Supplementary Fig. 12.

**Real-time RT-PCR.** RNA was isolated with a miRNeasy Mini kit (Qiagen) and cDNA was generated by reverse transcription (Applied Biosystems). Primers from Applied Biosystems and a Prism 7900HT (Applied Biosystems) were used for real-time PCR analysis of all genes; results are presented relative to U6 or Rpl13 expression (PCR primers are listed in Supplementary Table 1).

**Immunoblot analysis.** Cells were analyzed by immunoblot 72 h after transduction. Proteins were separated by 4–20% SDS-PAGE, followed by standard immunoblot analysis. Primary and secondary antibodies are listed in Supplementary Table 2.

**Retroviral vector construction and retrovirus production.** Phf19 cDNA or Phf19mut cDNA and Thy1.1 linked by sequence encoding the picornavirus 2 A ribosomal skip peptide was cloned together into the MSGV-1 vector. Phf19ShRNAF and Phf19ShRNAR primers, or Ship1 gRNAF and Ship1 gRNAR primers, respectively, were annealed and cloned into a modified pMKO.1 GFP vector. Platinum-E cell lines (Cell Biolabs) were used for gamma-retroviral pro-duction by transfection with DNA plasmids through the use of Lipofectamine 2000 (Invitrogen) and collection of virus 40 h after transfection (shRNA and gRNA primers are listed in Supplementary Table 1).

**In vitro activation and transduction of CD8$^+$ T cells.** CD8$^+$ T cells were separated from non-CD8$^+$ T cells with a MACS-negative selection kit (Miltenyi Biotech) and were activated on plates coated with anti-CD3ε (2 μg/ml; BD Bios-ciences) and soluble anti-CD28 (1 μg/ml; BD Biosciences) in culture medium containing rhIL-2 (120 IU/ml; Chiron). Negatively enriched, naïve CD44$^-$ CD62L$^+$ CD8$^+$ cells (Miltenyi Biotech) were employed in experiments utilizing Ezh2, Jarid2, and Phf19-deficient T cells. Virus was 'spin-inoculated' at 2000 × g for 2 h at 32 °C onto plates coated with retronectin (Takara). CD8$^+$ T cells activated for 24 h were transduced following standard protocol.

**Adoptive cell transfer, infection, and tumor treatment.** Cells ($10^5$–$4 × 10^6$ cells) were adoptively transferred into hosts followed by infection with $2 × 10^7$ pfu of recombinant vaccinia virus expressing human gp100 (Virapur) together with the indicated combination of exogenous cytokines ($2.4 × 10^5$ IU/dose of rhIL-2 for 6 doses every 12 h). Female C57BL/6 mice were injected subcutaneously with $3 × 10^5$ B16 or B16 (H-2$^b$)-hgp100 melanoma cells. Calipers were used to measure tumors, and the products of the perpendicular diameters were recorded. All experiments were performed in a blinded, randomized fashion thrice  weekly. Mice were euthanized by $CO_2$ or cervical dislocation when tumor size approached 2 cm in diameter, or impeded their movement.

**Enumeration of adoptively transferred cells.** Mice were euthanized after infection at indicated time points by cervical dislocation. Splenocytes or tumor infiltrating lymphocytes were counted by trypan blue exclusion. The frequency of transferred T cells was determined by measurement of the expression of CD8 and GFP, or Thy1.1, by flow cytometry. The absolute number of pmel-1 cells was calculated by multiplication of the total cell count with the percentage of CD8$^+$GFP$^+$ or CD8$^+$Thy1.1$^+$ cells.

**Pharmacological inhibition of Akt signaling.** miR-155 or Ctrl-miR-overexpressing cells were sorted 5 days after transfer of $6 × 10^5$ miR-155 or Ctrl-miR-overexpressing cells into wild-type mice in conjunction with gp100-VV administration. After sorted, $5 × 10^5$ cells per condition were incubated in vitro for 6 h in the presence or absence of 1 μmol/L Akt inhibitor VIII (Calbiochem) before downstream processing.

**Nanostring.** Cells were sorted ex vivo and total RNA was isolated with a miRNeasy Mini kit (Qiagen). Ten nanogram total RNA was used for Nanostring analysis following the NanoString nCounter Expression CodeSet Design Manual. Back-ground levels were calculated and subtracted from the samples, which were then normalized against the positive control and housekeeping gene probes. Expression heat maps were generated with the R package ComplexHeatmap[59].

**RNA-seq.** RNA concentration was determined with the Qubit RNA broad range assay in the Qubit Fluorometer (Invitrogen) and RNA integrity was determined with Eukaryote Total RNA Nano Series II ChIP on a 2100 Bioanalyzer (Agilent). RNA-seq libraries were prepared from 4 μg of total RNA via the TruSeq RNA sample prep kit according to manufacturer's protocol (Illumina). In brief, oligo-dT purified mRNA was fragmented and subjected to first and second strand cDNA synthesis. cDNA fragments were blunt-ended, ligated to Illumina adaptors, and PCR amplified to enrich for the fragments ligated to adaptors. The resulting cDNA libraries were verified and quantified on Agilent Bioanalyzer and paired-end sequencing was carried out on the GAIIx Genome Analyzer (Illumina).

RNA-seq analyses were performed using ≥2 biological replicates. Briefly, total RNA was prepared from cells using the RNeasy Plus Mini Kit (Qiagen). Two hundred nanogram total RNA was subsequently used to prepare RNA-seq library by using TruSeq RNA sample prep kit (FC-122–1001, Illumina) according to the manufacturer's instructions. Raw sequence reads were aligned to the mouse genome (NCBI37/mm9) with TopHat 2.1.1, and uniquely mapped reads were used to calculate gene expression[60]. The mouse reference genome sequences and the annotation (mm9) were downloaded from the UCSC genome browser for RNA-seq analysis. RNA-seq reads that fell on transcripts of each gene were counted, and differentially expressed genes were identified with the Bioconductor packages edgeR[61] or DESeq2[62]. Differentially expressed genes (DEGs) were required to meet to the criteria: over 1.8-fold change difference, and false discovery rate (FDR) <0.05. Fold change heat maps were generated with the Bioconductor ComplexHeatmap[59].

**Gene-set enrichment analysis of RNA-seq data.** Mouse gene symbols were first mapped to the orthologous human genes using the homology information available from the MGI website (ftp://ftp.informatics.jax.org/pub/reports/HMD_HGNC_Accession.rpt) and were ranked by the fold changes of the gene expression as profiled by RNA-seq. Then, gene-set enrichment was analyzed using GSEA 3.0 software (http://software.broadinstitute.org/gsea/downloads.jsp)[63]. Pathway Analysis was performed on the identified differentially expressed genes list using the Core Analysis function included in Ingenuity Pathway Analysis (IPA, Qiagen).

**ChIP-seq and ChIP-qPCR.** T cells were crosslinked with 2% paraformaldehyde at room temperature for 10 min and lysed in Farnham buffer (5 mM PIPES pH 8.0; 85 mM KCl; 0.5% NP-40) and subsequently in RIPA buffer (1x PBS; 1% NP-40; 0.5% sodium deoxycholate; 0.1% SDS). Sheared chromatin was immunopreci-tated with anti-H3K27me3 antibody (Millipore, 07–449) and washed successively in buffer I (20 mM Tris HCl pH 8.0, 150 mM NaCl, 2 mM EDTA, 0.1% SDS, 1% Triton X-100); buffer II (20 mM Tris HCl pH 8.0, 500 mM NaCl, 2 mM EDTA, 0.1% SDS, 1% Triton X-100); three times of buffer III (100 mM Tris HCl pH 8.0, 500 mM LiCl, 1% NP-40; 1% sodium deoxycholate). For sequencing of immuno-precipitated DNA, DNA fragments were blunt-end ligated to Illumina adaptors,

amplified, and sequenced by using the HiSeq 2000 platform (Illumina). Single-end reads of 50 bp were obtained by using the Illumina Pipeline. Sequenced reads were aligned to the mouse genome (NCBI37/mm9) with Bowtie 2.2.8; only uniquely mapped reads were retained. H3K27me3 enriched regions were detected using SICER 1.1 algorithm[64], and the window size, gap size, and FDR were set to 200 bp, 600 bp, and 5%, respectively. Genomic graphs were generated and viewed using the IGV 2.3.92 (Integrative Genomics Viewer). For ChIP-qPCR immunoprecipitated DNA, DNA fragments were amplified for selected transcription factors using the primers listed in Supplementary Table 1.

**Statistical analyses**. Statistical analysis was performed using Prism software (Graph Pad). Unpaired, two-tailed Student's t-tests were used for comparison of data such as gene expression levels, cell proliferation and functionality (numbers and percentage), and tumor growth slopes. A Log-rank (Mantel-Cox) Test was used for comparison of survival curves.

**Reporting summary**. Further information on research design is available in the Nature Research Reporting Summary linked to this article.

## Data availability

The ChIP-seq, RNA-seq, and Nanostring data are available in the Gene Expression Omnibus (GEO) (https://www.ncbi.nlm.nih.gov/geo/query/acc.cgi) under accession numbers GSE99918, GSE99801 and GSE128664. The source data underlying Figs. 1b–f; 2a–f; 3b–h; 4b–d, f–i; 5a–e; 6f, g; 7b–i and Supplementary Figs. 2d, f; 4b, e; 5b; 6b; 7a–d; 8a; 9b; 10b; 11a, b are provided as a Source Data file. All other data are available from the authors upon reasonable requests.

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

## Acknowledgements

This research was supported by the Intramural Research Programs of the US National Institutes of Health, National Cancer Institute (ZIABC011480), National Institute of Arthritis, Musculoskeletal and Skin Diseases, and National Institute of Allergy and Infectious Diseases. We thank K. Hanada for providing B16 (H-2$^b$)-hgp100. This work used the computational resources of the NIH HPC Biowulf cluster (http://hpc.nih.gov).

## Author contributions

Y.J., J.F., H.W., T.W., J.H., S.G., N.L., J.B.L., X.Y., J.D.H., T.E., S.D., N.V.H., V.K. and W.G.T. performed experiments; Y.J., J.F., W.Z., and L.G. analyzed experiments; Y.J., J.F., H.W., V.S. and L.G. designed experiments; T.W., S.H., L.D.C., Y.Z. and S.M. contributed reagents; J.F., J.B.L., L.D.C., and Y.Z. edited the manuscript; Y.J. and L.G. wrote the manuscript.

## Additional information

**Competing interests:** Y.J. and L.G. are the inventors of an international pending patent on Phf19 technology ("T-CELLS MODIFIED TO OVEREXPRESS PHF19" PCT/US2018/036125 filed on 05 June 2018, applied by THE UNITED STATES OF AMERICA, AS REPRESENTED BY THE SECRETARY, DEPARTMENT OF HEALTH AND HUMAN SERVICE; Office of Technology Transfer National Institutes of Health 6011 Executive Boulevard, Suite 325, MSC 7660 Bethesda, Maryland 20892–7660, US). The remaining authors declare no competing interests.

