## [Peer Review File · Nature Communications]

Reviewers' comments:

Reviewer #1 (T activation/tolerance/energy, tumor immunity)(Remarks to the Author):

In the present study Yun Ji et al present their findings regarding the role of miR-155 to potentiate T cell responses through epigenetic mechanisms. The authors support the conclusion that miR-155 prevents the differentiation of T effector cells thereby maintaining a more stem-cell like phenotype by regulating the function of PRC2 complex. They found that Ezh2, the key enzymatic component of the PRC2 complex was required for this function because the effect of miR-155 to prevent the terminal differentiation of T effector cells was diminished in the absence of Ezh2. They identified Phf19, a mediator of PRC2 recruitment to histones, as a key target of miR-155 that might be responsible for these effects on T cell effector differentiation because Phf19 was upregulated in miR-155 overexpressing cells and was reduced in miR-155 deficient T cells. This increase was secondary to miR-155-mediated activation of Akt due to downregulation of the Akt inhibitor, Ship1. Because modifications of these signaling targets of miR-155 alter T effector cell function, the authors propose that this pathway might be a potential therapeutic target for improvement of cancer immunotherapy. The results are interesting and identify one more among the numerous targets of miR-155. The studies were well designed but the data are not fully supportive of the authors conclusions and interpretations.

Specific points:

1. The study has focused on the effects of miR-155-Phf19-PRC2 in CD8+ T effector cells. However, the PRC2 complex has an important role in CD4+ and Treg cells, which are also important regulators of anti-tumor immunity. In order to understand the biological relevance of the miR-155-Phf19-PRC2 axis and its potential role in immunotherapy, the effects of this pathway in CD4+ and Treg cells should also be examined.
2. The author's model that miR-155 prevents T effector cell differentiation does not reconcile with their data that miR-155 overexpressing and Phf19 overexpressing cells produce more effector cytokines.
3. Figure 2: What is the reason of the differential impact of miR-155 on cytokine production and polyfunctionality in early vs. later time points in Ezh2^{-/-} cells whereas no such effect is observed in Ezh2^{+/+} cells? Mechanistic understanding of these findings is required.
4. In the same context, it is important to determine if other mechanisms related to T effector and memory differentiation such as survival, apoptosis and autophagy are affected by miR-155. Without delineating these issues that have important roles in T effector cell differentiation and expansion, it is not possible to state that the specific mechanism by which miR-155 affects differentiation and expansion of CD8+ T effector cells is mediated via Ship1-Akt-Phf19.
5. Figure 4d and g: The effect of miR-155 and Ship1 sgRNA on Akt phosphorylation is minimal and non-compelling as the central mechanism that drives all the molecular, cellular and functional outcomes of miR-155 reported in the study. This is the central point of the study and falls short to support the authors' conclusions and proposed model.
6. Figure 5: In panel b, complete kinetics of Ezh2 relative expression in the acute response and the transition to memory phase should be assessed as in panel a. This is important because it based on the results of Figure 2d, one can conclude that Ezh2 itself plays a different role in the early vs. the later phase of the immune response.
7. Figure 8: The model does not reflect the authors' findings on the role of Jarid2 downstream of miR-155. The model presents a similar effect of miR-155 on Jarid2 and Ship1, albeit to a different magnitude, whereas the data show that Jarid2 is not involved (or might have opposing effect) in miR-155 mediated outcomes in CD8+ T cells.
8. The title of the paper is not appropriate. The study did not show that harnessing Phf19 potentiates cancer immunotherapy. Prolonging the survival of mice in a lethal cancer model is not representing cancer immunotherapy. The title should accurately reflect the findings of the study.

Reviewer #2 (CD8, T exhaustion, viral infection)(Remarks to the Author):

Mir155 is an important micro RNA regulating T cell differentiation and survival. It is up-regulated in effector and exhausted T cells. Many downstream targets are predicted to be regulated by mir-115. Here, Yu et al interrogate the epigenetic regulation through mir-155 in T cells and report that it can enhance T cell expansion by enhancing PRC2 activity in an Ezh2 dependent fashion indirectly, through disinhibition of Akt signaling. Jarid2, a mir155 target and regulator of PRC2 was analyzed but no significant impact of Jarid2 deficiency on T cell expansion was identified, with small differences in effector differentiation and tumor surveillance by Jarid2 ko T cells. In contrast, the authors identify transcription factor Phf19 thought to interact with PRC2 as being regulated downstream of mir155-Akt signaling. Comparison of gene targets silenced by Phf19 and targeted by mir155 overexpression show a significant overlap. Phf19 ko T cells show significantly worse expansion, and overexpression of mir155 in Phf19 ko T cells couldnt rescue this deficiency, suggesting that Phf19 is a relevant effector downstream of mir155. Through mutation of chromatin binding sites, the authors identify chromatin binding as an important property of Phf19 action that is also required for T cell anti-tumor effects in a B16 melanoma model.

This is a study investigating downstream pathways of an important microRNA in their ability to modulate the T cell response against infection and cancer. Many aspects of the study are well controlled for, and the main conclusions about a link of mir155 overexpression affecting PRC2 activity via Phf19 are supported by the data presented.

However, the complex pathway suggested is not completely validated. A key question given the indirect connection proposed that is not answered in this study is whether pharmacologic Akt inhibition abrogates the effects of mir155 OE on Phf19? Another is whether Ezh2 inhibition abrogates the stimulatory effects of phf19 on T cell expansion and anti-tumor response in the mir155oe setting. These questions should be addressed prior to publication.

Additional comments

Figure 1a – The authors write that (line 88following) ... mir155 oe cells showed reduced expression of genes silenced by PRC2 in mir 155. However, it is confusingly represented in the GSEA plots which are typically displaying the expression of gene expression across groups. Here, it appears opposite in figure 1a to me since the GSEA plot as interpreted by conventional use of this technique would show enrichment of all genesets in 1a including PRC2 targets in mir155oe over control – this is probably due to use of pre-calculated fold change metrics, please check and I would suggest to use the conventional display or at least explain better. This also affects figure 6d.

Figure 1h – the depth of analysis regarding the epigenetic regulation on the level of T cell exhaustion or the key phenotypic markers measured for assessing differentiation discussed in the paper is relatively weak. For example, data on H3K27me3 levels regulating Klrp1, CD62L and Pd-1 expression would be of interest here.

Figure 3b – the immunoblot is hard to judge. Please quantify expression relative to Actin control

Figure 6 Gene expression was tested in CD62L-KLRG1- cells at peak of immune response. I understand the rationale for studying this subset thought to be less committed to SLEC or MPEC and pending differentiation decisions but were these subsets also studied separately? Where there differences in Phf19 ko cells already at the naive stage?

Figure 6c Please explain in manuscript why those gene sets studied were selected. In figure 6c there is no match between the numbers given and the dots indicated.

The authors suggest that mir155 could be therapeutically harnessed to counter exhaustion. However it is already strongly induced in exhausted cells and, if overexpressed, promotes terminal differentiation

(Stelekati et al Cell Reports 2018). The authors also discuss the differences in their study and results by others with regards to the role of Ezh2 on T cell differentiation and indicate differences could be due to the system studied here which includes knockout cells already at the naïve stage. If the authors feel that the dominant effects observed may be due to modulation of very early T cell differentiation, possibly at the first cell division (p10), this might not adequately translate to a therapeutic scenario where T cells are already differentiated.

Response to referees

We would like to thank the reviewers for their thoughtful and very constructive feedback regarding our manuscript, **“miR-155 harnesses Phf19 to potentiate cancer immunotherapy through epigenetic reprogramming of T cell fate”**

In response to their queries, we have incorporated new experimental data into our submission which we believe substantially improves the strength of our conclusions. A complete point-by-point response to specific reviewer comments is detailed below.

We hope that we have provided satisfactory answers to the reviewer’s comments and that you will find our revised manuscript suitable for publication in *Nature Communications*.

Reviewer #1

1. *“The study has focused on the effects of miR-155-Phf19-PRC2 in CD8⁺ T effector cells. However, the PRC2 complex has an important role in CD4⁺ and Treg cells, which are also important regulators of anti-tumor immunity. In order to understand the biological relevance of the miR-155-Phf19-PRC2 axis and its potential role in immunotherapy, the effects of this pathway in CD4⁺ and Treg cells should also be examined.”*

Understanding the function of the miR-155-Phf19-PRC2 axis in CD4⁺ T cells as well as in other immune cells is certainly important, but we feel that goes beyond the scope of our current work, which is centered around the biological and therapeutic effects of this axis in CD8⁺ T cells in the setting of adoptive T cell therapy. Studying the biological relevance of the miR-155-Phf19-PRC2 axis in CD4⁺ T cells and Tregs will not affect the conclusions of our study since they are based on experiments in which each component of the axis is exclusively altered on transferred CD8⁺ T cells.

2. *“The author’s model that miR-155 prevents T effector cell differentiation does not reconcile with their data that miR-155 overexpressing and Phf19 overexpressing cells produce more effector cytokines.”*

It is well established that as CD8⁺ T cells progressively differentiate into terminal effector cells, they first lose the capacity to produce interleukin-2 (IL-2) and then tumor necrosis factor (TNF), before ultimately becoming monofunctional interferon (IFN)- γ producers (see cartoon below) or functionally exhausted¹.

[Redacted]

Editorial Figure 1. Functional changes during CD8⁺ T cell differentiation. Adapted from Appay *et al.* Cytometry A , 2008¹.

Thus, our findings that miR-155 and Phf19 overexpressing cells produce more cytokines are consistent with our conclusions that the miR-155-Phf19-PRC2 axis limits CD8⁺ T cell terminal effector differentiation and senescence.

3. "Figure 2: What is the reason of the differential impact of miR-155 on cytokine production and polyfunctionality in early vs. later time points in Ezh2^{-/-} cells whereas no such effect is observed in Ezh2^{+/+} cells? Mechanistic understanding of these findings is required."

We have previously shown that overexpression of miR-155 enables CD8⁺ T cells to sustain cytokine production at later time points after transfer (Ji *et al.* PNAS 2015²). In this current work, we show that this phenomenon depends on the presence of Ezh2 and PRC2 function. As discussed above, CD8⁺ T cell polyfunctionality is a measure of their differentiation status. As shown in Fig. 2b,c, the frequency of KLRG1⁺ terminal effectors becomes considerably higher in miR-155 Ezh2^{-/-} compared to miR-155 Ezh2^{+/+} cells only at later time points. Thus, by limiting CD8⁺ T cell terminal effector differentiation *via* PRC2-mediated silencing of pro-differentiating TF, miR-155 allows for sustained cytokine production at later time points.

4. "In the same context, it is important to determine if other mechanisms related to T effector and memory differentiation such as survival, apoptosis and autophagy are affected by miR-155. Without delineating these issues that have important roles in T effector cell differentiation and expansion, it is not possible to state that the specific mechanism by which miR-155 affects differentiation and expansion of CD8⁺ T effector cells is mediated via Ship1-Akt-Phf19."

We have previously reported that miR-155-deficient CD8⁺ T cells have defects in cell accumulation due to impaired cell proliferation and increased apoptosis (Dudda *et al.* Immunity 2013³). At the reviewer request's request, we have now examined the effect of miR-155 overexpression on CD8⁺ T cell survival and apoptosis. First, we have re-analyzed our RNA-seq dataset on ex vivo-isolated miR-155 and Ctrl-miR CD8⁺ T cells using the Ingenuity Pathway Analysis software to determine whether miR-155 influenced the expression of gene networks known to regulate cell survival and proliferation. We found that miR-155 overexpression orchestrated several pathways regulating cell proliferation and death. The top-scoring pathways identified by the IPA software were cell cycle progression, DNA replication, M phase, DNA repair, checkpoint control, DNA metabolism, cell proliferation and cell death. increased cell proliferation and reduced cell apoptosis. Consistent with this gene signature, miR-155 CD8⁺ T cells exhibited increased

proliferation (BrdU incorporation) and reduced apoptosis (Annexin V). These new data are shown below and in a new **Supplementary Fig. 2**.

Supplementary Fig. 2. miR-155 enhances CD8⁺ T cell proliferation and survival. **a** Top-scoring biological functions differentially regulated between miR-155 and Ctrl-miR pmel-1 CD8⁺ T cells 5 days after in vivo stimulation with gp100-VV identified with the IPA software. **b** Cell cycle, cellular assembly and organization, DNA replication, recombination and repair network. Upregulated genes are displayed in red. Solid and dashed lines between genes represent known direct and indirect gene interactions, respectively. **c-f** Flow cytometry analysis (**c**, **e**) and percentage (**d**, **f**) of BrdU incorporation (**c**, **d**) and Annexin V (**d**, **f**) into pmel-1 cells 5 days after adoptive transfer of 3 x 10⁵ miR-155 and Ctrl-miR pmel-1 CD8⁺ T cells into wild-type mice infected with gp100-VV.

To describe these data, we have modified the text as follows:

“As previously shown², CD8⁺ T cells overexpressing miR-155 expanded and survived better than controls (Fig. 1c, Supplementary Fig. 2)”

To determine if miR-155 regulates CD8⁺ T cell autophagy we measured the abundance of the microtubule-associated proteins LC3b-I and LC3b-II as well as the ubiquitin-binding scaffold protein p62 (sequestosome-1) in sorted miR-155 and Ctrl-miR CD8⁺ T cells. LC3b-II, the lipidated form of LC3b-I, is a classic marker of autophagosomes and is incorporated into the elongating membrane that eventually forms the autophagosome. p62 is a widely used marker for measuring autophagic activity, as it is both a substrate of autophagy and an adaptor in targeting ubiquitinated proteins for lysosomal degradation via the autophagy pathway. We did not find significant differences in both LC3b (LC3b-I and LC3b-II) and p62 protein accumulation between miR-155-overexpressing cells and controls, indicating that miR-155 does not regulate autophagy in CD8⁺ T cells. These data are provided below for editorial use.

Editorial Fig. 2. miR-155 does not regulate autophagy. Autophagy was evaluated *ex vivo* by immunoblot of LC3b-I/LC3b-II and p62 protein levels in miR-155 or Ctrl-miR-overexpressing cells sorted 3 and 5 days following the adoptive transfer of 3 x 10⁵ miR-155 or Ctrl-miR-overexpressing cells into wild-type mice in conjunction with gp100-VV administration.

5. “Figure 4d and g: The effect of miR-155 and Ship1 sgRNA on Akt phosphorylation is minimal and non-compelling as the central mechanism that drives all the molecular, cellular and functional outcomes of miR-155 reported in the study. This is the central point of the study and falls short to support the authors’ conclusions and proposed model.”

We respectfully disagree with the reviewer's opinion. Both miR-155 and Ship1 sgRNA causes a clear increase of pAkt. We previously showed a more striking blot describing the regulatory influences of miR-155 on Ship1 and Akt phosphorylation (Ji *et al.* PNAS 2015², and below for editorial use). Because, both miR-155 regulation of Ship1 and Ship1 regulation of pAkt are well-established in the literature in many cell types including CD8⁺ T cells^{4, 5, 6}, we don't believe that a further optimization of the immunoblot is necessary to support our conclusions.

Editorial Fig. 3. miR-155 potentiates pAkt via downregulation of Ship1. Adapted from Ji *et al.* PNAS 2015².

6. "Figure 5: In panel b, complete kinetics of *Ezh2* relative expression in the acute response and the transition to memory phase should be assessed as in panel a. This is important because it based on the results of Figure 2d, one can conclude that *Ezh2* itself plays a different role in the early vs. the later phase of the immune response."

At the reviewer's request we have modified Figure 5a,b to include a complete kinetic of *Ezh2* and *Mir155* expression as shown below.

Fig. 5a. Coordinate expression of *Ezh2*, *Phf19*, and *Mir155* in CD8⁺ T cells responding to viral infection. Quantitative RT-PCR analysis of the expression of *Ezh2*, *Phf19* and *Mir155* after transfer of 10⁵ pmel-1 CD8⁺Ly5.1⁺ T cells into wild-type mice in conjunction with gp100-VV administration assessed at the indicated points. *Ezh2* and *Phf19* levels are relative to *Rpl13*, *Mir155* levels are relative to *U6*.

These findings are described in text as follows:

"We found that *Phf19* was strongly induced at the early stages of acute immune response, sharply downregulated at peak effector response and maintained at low levels throughout transition to memory phase (Fig. 5a). These findings suggested a potential role of *Phf19* in regulating CD8⁺ T cell effector differentiation. The rapid spike of induction was similarly observed for *Ezh2*, indicating coordinated expression of PRC2 and its associated factor *Phf19* during the immune response (Fig. 5a). Remarkably, *Mir155* followed a virtually identical pattern of expression, emphasizing the interplay between these molecules during physiologic immune response (Fig. 5a)."

7. "Figure 8: The model does not reflect the authors' findings on the role of *Jarid2* downstream of miR-155. The model presents a similar effect of miR-155 on *Jarid2* and *Ship1*, albeit to a different magnitude,

whereas the data show that Jarid2 is not involved (or might have opposing effect) in miR-155 mediated outcomes in CD8+ T cells.”

The model in Fig. 8 correctly summarizes the results shown in the manuscript. We have shown that miR-155 downregulates both Ship1 (Fig. 4d and Editorial Fig. 3) and Jarid2 (Fig. 3b). In our settings, Jarid2 is a weak positive regulator (slim arrow) of PRC2 function. Thus, downregulation of Jarid2 results in a mild increase in terminal effectors (Fig. 3g) and a small impairment of anti-tumor function (Fig. 4h). The downregulation of Ship1 results in an increase in Phf19 via Akt signaling (Fig. 4) which has dominant effect in our model on PRC2 activity (thick arrow). Thus, despite the inhibition of Jarid2, the overall effect of miR-155 is an enhancement of PRC2 function via Phf19, which results in reduced terminal differentiation and better anti-tumor responses.

8. “The title of the paper is not appropriate. The study did not show that harnessing PhF19 potentiates cancer immunotherapy. Prolonging the survival of mice in a lethal cancer model is not representing cancer immunotherapy. The title should accurately reflect the findings of the study.”

By *Nature* journal’s definition (<https://www.nature.com/subjects/cancer-immunotherapy>) cancer immunotherapy “is a therapy used to treat cancer patients that involves or uses components of the immune system. Some cancer immunotherapies consist of antibodies that bind to, and inhibit the function of, proteins expressed by cancer cells. Other cancer immunotherapies include vaccines and T cell infusions.” The latter –the infusion of tumor-specific CD8⁺ T cells– is precisely what we have done in this study to induce tumor regression in tumor-bearing mice. Thus, we believe that the title of the paper is accurate since adoptive T cell transfer is indeed a form of cancer immunotherapy.

Reviewer #2

1. “A key question given the indirect connection proposed that is not answered in this study is whether pharmacologic Akt inhibition abrogates the effects of mir155 OE on Phf19?”

To determine whether pharmacologic inhibition of Akt diminishes the effects of miR-155 overexpression on Phf19 we cultured miR-155 and Ctrl-miR CD8⁺ T cells sorted 5 days after transfer into mice infected with gp100-VV in the presence of AKT inhibitor VIII^{7, 8} and measured *Phf19* expression after a 6h incubation. Consistent with our previous data, inhibition of Akt signaling significantly reduced the mRNA levels of *Phf19*. These findings are shown below and in a new panel Fig.4f.

Fig. 4f. miR-155 induces *Phf19* expression via Akt signaling. Quantitative RT-PCR analysis of the expression of *Phf19* mRNA in *ex vivo* sorted CD8⁺ T cells overexpressing miR-155 and Ctrl-miR after a 6h incubation with or without AKT inhibitor VIII (Akti).

To describe these findings, we have modified the manuscript as follows:

“To determine whether miR-155 induced *Phf19* by enhancing Akt signaling, we first measured *Phf19* expression in miR-155 and Ctrl-miR-overexpressing cells after incubation with the AKT inhibitor VIII or

transduction with a constitutively active form of Akt (AktCA)². Consistent with our prior findings, miR-155 upregulated *Phf19* (Fig. 4f, g). Strikingly, blockade of AKT signaling significantly inhibited *Phf19* expression (Fig. 4f) whereas constitutive Akt signaling drove *Phf19* expression to saturation, abrogating any further upregulation of *Phf19* by miR-155 (Fig. 4g)”

2. “Another is whether *Ezh2* inhibition abrogates the stimulatory effects of *Phf19* on T cell expansion and anti-tumor response in the *mir155oe* setting.”

We have shown in our original submission that miR-155 transduced CD8⁺ T cells despite expressing heightened levels of *Phf19* (Fig. 4a,b) have impaired expansion (Fig. 2a) and antitumor function (Fig. 2e,f) in the genetic absence of *Ezh2*. To determine if overexpression of *Phf19* can overcome *Ezh2* deletion, we tested the antitumor responses of *Ezh2*-deficient CD8⁺ T cells transduced with *Phf19*Thy1.1 or Thy1.1 control. Consistent with the notion that *Ezh2* and PRC2 are downstream of *Phf19*^{9, 10}, we found that overexpression of *Phf19* could not rescue the antitumor responses of *Ezh2*-deficient cells. These results are shown below and in the new Supplementary Fig. 10.

Supplementary Fig. 10. *Ezh2* is required for the enhanced CD8⁺ T cell antitumor immunity conferred by *Phf19*. **a, b** Tumor size (mean \pm s.e.m.) (**a**) and survival curve (**b**) of B16 tumor-bearing mice after adoptive transfer of 4×10^5 pmel-1 *Ezh2*^{+/+} or *Ezh2*^{-/-} CD8⁺ T cells transduced with *Phf19*Thy1.1 or Thy1.1 in conjunction with gp100-VV and IL-2 administration.

To describe these findings, we have modified the manuscript as follows:

“This conclusion was further strengthened by the finding that the therapeutic impact of *Phf19* overexpression was abolished in CD8⁺ T cells lacking *Ezh2* (Supplementary Fig. 10). Altogether these findings demonstrate that *Phf19* potentiates CD8⁺ T cell anti-tumor activity by epigenetic reprogramming via PRC2.”

Reviewer #2 Minor comments

1. “Figure 1a – The authors write that (line 88following) ... *mir155 oe* cells showed reduced expression of genes silenced by PRC2 in *mir-155*. However, it is confusingly represented in the GSEA plots which are typically displaying the expression of gene expression across groups. Here, it appears opposite in figure 1a to me since the GSEA plot as interpreted by conventional use of this technique would show enrichment of all gene sets in 1a including PRC2 targets in *mir155oe* over control – this is probably due to use of pre-calculated fold change metrics, please check and I would suggest to use the conventional display or at least explain better. This also affects figure 6d.”

Figure 1a shows GSEA plots of genes that are targeted for silencing by PRC2 in stem cells and progenitor cells. These gene sets are negatively enriched (negative ES scores) in miR-155 overexpressing CD8⁺ T cells, which in other words means that miR-155 transduced cells express reduced amounts of these transcripts compared to Ctrl-miR cells. The same concept applies to the bottom panels of Figure 6d, which

demonstrates a negative enrichment of genes (i.e. lower expression, negative ES scores) in *Phf19*-deficient cells compared to WT cells. Conversely, a positive enrichment (i.e. higher expression, positive ES scores) in miR-155 overexpressing cells versus Ctrl-miR is shown in the upper panels.

2. *“Figure 1h – the depth of analysis regarding the epigenetic regulation on the level of T cell exhaustion or the key phenotypic markers measured for assessing differentiation discussed in the paper is relatively weak. For example, data on H3K27me3 levels regulating Klrp1, CD62L and Pd-1 expression would be of interest here.”*

At the reviewer’s request we have analyzed the H3K27me3 levels of *Klrp1*, CD62L (*Sell*) and PD1 (*Pdcd1*) in miR-155 and Ctrl-miR KLRG1⁻ CD8⁺ T cells. We did not observe any deposition of H3K27me3 at the promoters of these genes in miR-155 transduced cells indicating that these molecules are not under the epigenetic suppression mediated by PRC2 in these cells. *Sell* is one of the few genes (194) which exhibit increased H3K27me3 in Ctrl-miR compared to miR-155 overexpressing cells. We provide these results below for editorial use since they are not critical for the overall conclusions of the paper.

Editorial Fig.4. H3K27me3 marks and RNA-seq reads at *Pdcd1*, *Sell* and *Klrp1* loci in miR-155 and Ctrl-miR-overexpressing cells. ChIP-seq was performed on Non T_E KLRG1⁻ pmel-1 CD8⁺ T cells transduced with miR-155 or Ctrl-miR for 5 days. Gene expression was evaluated by RNA-seq of KLRG1⁻CD62L⁻ cells sorted from transferred cells 5 days after adoptive transfer of 3 x 10⁵ miR-155 or Ctrl-miR-overexpressing cells into wild-type mice in conjunction with gp100-VV administration.

3. *“Figure 3b – the immunoblot is hard to judge. Please quantify expression relative to Actin control.”*

We have quantified the expression of Jarid2 protein and confirmed that it is significantly downregulated in miR-155 overexpressing CD8⁺ T cells (-2.17 fold). It should be noted that this level of protein reduction is in line with the downregulation of Jarid2 transcripts (-2.78 fold) measured by RNA-seq (Supplementary Fig. 1).

4. *“Figure 6 Gene expression was tested in CD62L-KLRG1- cells at peak of immune response. I understand the rationale for studying this subset thought to be less committed to SLEC or MPEC and pending differentiation decisions but were these subsets also studied separately? Where there differences in Phf19 ko cells already at the naïve stage?”*

In addition to CD62L⁻KLRG1⁻ cells, we studied the gene expression in bulk and SLEC (KLRG1⁺CD62L⁻). As shown below, the gene expression profile of miR-155 overexpressing KLRG1⁻CD62L⁻ cells highly correlates with dataset obtained from bulk CD8⁺ T cells overexpressing miR-155 as well as from SLEC

(Editorial Fig. 5). These findings indicate that the differences between miR-155 overexpressing cells and controls are mainly driven by the activity of miR-155 rather than a skewing in T cell subset distribution.

Editorial Fig.5. The gene expression profile of miR-155 overexpressing cells is driven by miR-155 levels rather than a skewed subset distribution. Correlation of differentially expressed genes between miR-155 overexpressing and Ctrl-miR cells obtained from bulk CD8⁺ T cells and KLRG1⁻CD62L⁻ sorted cells (left panel) and from KLRG1⁺CD62L⁻ and KLRG1⁻CD62L⁻ cells (right panel). r represents the correlation score of Pearson's product-moment correlation test.

We did not study MPEC nor the gene expression profile of Phf19 KO cells at a naïve stage.

5. "Figure 6c Please explain in manuscript why those gene sets studied were selected. In figure 6c there is no match between the numbers given and the dots indicated."

We selected all gene sets that were significantly enriched in Phf19KO vs WT cells (FDR < 0.25). An FDR of 25% indicates that the result is likely to be valid 3 out of 4 times, is conventionally employed in the setting of exploratory discovery where one is interested in finding candidate hypothesis to be further validated as a results of future research. Given the lack of coherence in most expression datasets and the relatively small number of gene sets being analyzed, using a more stringent FDR cutoff may lead to overlook potentially significant results. We have added this information in the main text (see http://software.broadinstitute.org/gsea/doc/GSEAUserGuideFrame.html?Interpreting_GSEA).

Figure 6c displays a parts of whole plot (Prism software, Graph Pad) where each dot represents 1% of the data set. The data set in the left panel represents a total $n=32$ whereas the right panel represents a total $n=1113$.

6. "The authors suggest that mir155 could be therapeutically harnessed to counter exhaustion. However, it is already strongly induced in exhausted cells and, if overexpressed, promotes terminal differentiation (Stelekati et al Cell Reports 2018)."

Consistent with Stelekati et al¹¹ (Cell Reports 2018, Fig.4b), we showed that overexpression of miR-155 reduces the formation of terminally differentiated KLRG1⁺ CD8⁺ T cells. By contrast, we did not measure any defect in cytokine production nor upregulation of exhaustion molecules in our experimental setting which is based on an acute vaccination (vaccinia virus) rather than a chronic infection (LCMV cl13). It should also be noted that while miR-155 overexpression enhances the proliferation and survival of terminally exhausted cells in chronic infection, it still preserves the TCF1^{high} progenitor pool¹¹, which is critical to sustains long-term antiviral immunity^{12, 13, 14}. Thus, we do not believe that overexpressing miR-155 in antitumor CD8⁺ T cells will be detrimental. Indeed, the results of our tumor experiments clearly exemplify how miR-155 overexpression can be harnessed therapeutically to potentiate the antitumor responses of CD8⁺ T cells.

7. "The authors also discuss the differences in their study and results by others with regards to the role of Ezh2 on T cell differentiation and indicate differences could be due to the system studied here which includes knockout cells already at the naïve stage. If the authors feel that the dominant effects observed

may be due to modulation of very early T cell differentiation, possibly at the first cell division (p10), this might not adequately translate to a therapeutic scenario where T cells are already differentiated."

The reviewer's interpretation that the dominant effects on PRC2/Ezh2 that we observe are due to modulation at early stage of T cell differentiation is correct. This, however, doesn't preclude the potential translation of this approach: thanks to advancement in clinical-grade cell sorting technology and CAR and TCR therapy it is now possible to confer antitumor reactivity to very early T cell subsets like T_{SCM} cells¹⁵.

We hope you find our paper acceptable for publication in *Nature Communications*.

Sincerely,

REFERENCES

1. Appay V, van Lier RA, Sallusto F, Roederer M. Phenotype and function of human T lymphocyte subsets: consensus and issues. *Cytometry A* **73**, 975-983 (2008).
2. Ji Y, *et al.* miR-155 augments CD8+ T-cell antitumor activity in lymphoreplete hosts by enhancing responsiveness to homeostatic gamma_c cytokines. *Proceedings of the National Academy of Sciences of the United States of America* **112**, 476-481 (2015).
3. Dudda JC, *et al.* MicroRNA-155 Is Required for Effector CD8(+) T Cell Responses to Virus Infection and Cancer. *Immunity* **38**, 742-753 (2013).
4. O'Connell RM, Chaudhuri AA, Rao DS, Baltimore D. Inositol phosphatase SHIP1 is a primary target of miR-155. *Proceedings of the National Academy of Sciences of the United States of America* **106**, 7113-7118 (2009).
5. Yamanaka Y, *et al.* Aberrant overexpression of microRNAs activate AKT signaling via down-regulation of tumor suppressors in natural killer-cell lymphoma/leukemia. *Blood* **114**, 3265-3275 (2009).
6. Lind EF, Elford AR, Ohashi PS. Micro-RNA 155 is required for optimal CD8+ T cell responses to acute viral and intracellular bacterial challenges. *J Immunol* **190**, 1210-1216 (2013).
7. Crompton JG, *et al.* Akt inhibition enhances expansion of potent tumor-specific lymphocytes with memory cell characteristics. *Cancer research* **75**, 296-305 (2015).
8. van der Waart AB, *et al.* Inhibition of Akt signaling promotes the generation of superior tumor-reactive T cells for adoptive immunotherapy. *Blood* **124**, 3490-3500 (2014).
9. Ballare C, *et al.* Phf19 links methylated Lys36 of histone H3 to regulation of Polycomb activity. *Nat Struct Mol Biol* **19**, 1257-1265 (2012).
10. Brien GL, *et al.* Polycomb PHF19 binds H3K36me3 and recruits PRC2 and demethylase NO66 to embryonic stem cell genes during differentiation. *Nat Struct Mol Biol* **19**, 1273-1281 (2012).
11. Stelekati E, *et al.* Long-Term Persistence of Exhausted CD8 T Cells in Chronic Infection Is Regulated by MicroRNA-155. *Cell reports* **23**, 2142-2156 (2018).
12. Wu T, *et al.* The TCF1-Bcl6 axis counteracts type I interferon to repress exhaustion and maintain T cell stemness. *Sci Immunol* **1**, (2016).
13. Im SJ, *et al.* Defining CD8+ T cells that provide the proliferative burst after PD-1 therapy. *Nature* **537**, 417-421 (2016).
14. Utzschneider DT, *et al.* T Cell Factor 1-Expressing Memory-like CD8(+) T Cells Sustain the Immune Response to Chronic Viral Infections. *Immunity* **45**, 415-427 (2016).

15. Sabatino M, *et al.* Generation of clinical-grade CD19-specific CAR-modified CD8+ memory stem cells for the treatment of human B-cell malignancies. *Blood* **128**, 519-528 (2016).

Reviewers' comments:

Reviewer #1 (Remarks to the Author):

In the revised manuscript the authors have performed additional work and have properly addressed most of the important issues of the first version. However, their response and approach regarding the non-compelling data on the effect of miR155 on Akt phosphorylation in Figure 4d is unsatisfactory. The authors answered that they have published a better blot in a previous manuscript published in PNAS. The fact that they had shown a better blot in a previous study (shown in editorial figure 3) does not mean that they should not provide a similar quality blot in the present manuscript. The readers of the present paper will not go back to the authors' previous publication to look at the better results published in that publication. An improved pAkt blot should be provided in Figure 4d.

Reviewer #2 (Remarks to the Author):

The authors have carefully addressed my concerns.

With respect to minor comment 1: The authors explain their GSEA depiction based off the rank ordered list. Their interpretation in the text is as follows 'Specifically, miR-155-overexpressing cells showed reduced expression of genes silenced by PRC2 in mouse and human embryonic stem cells (ESC) and progenitors^{18, 19} (Fig. 1a)'. While I agree with the statement reflecting the data it would be significantly more intuitive to present the data by a heatmap rather than a GSEA.

Reviewer #1

1. "In the revised manuscript the authors have performed additional work and have properly addressed most of the important issues of the first version. However, their response and approach regarding the non-compelling data on the effect of miR155 on Akt phosphorylation in Figure 4d is unsatisfactory. The authors answered that they have published a better blot in a previous manuscript published in PNAS. The fact that they had shown a better blot in a previous study (shown in editorial figure 3) does not mean that they should not provide a similar quality blot in the present manuscript. The readers of the present paper will not go back to the authors' previous publication to look at the better results published in that publication. An improved pAkt blot should be provided in Figure 4d."

At the reviewer request we have performed a new western blot analysis of pAkt in miR-155 and Ctrl-miR CD8⁺ T cells. This improved western blot clearly demonstrates that Akt signaling is enhanced in miR-155 transduced T cells (**Fig. 4d**). In addition, we have provided for editorial use a quantification of pAkt from four independent T cell cultures. These data show that the increase of pAkt is robust and highly reproducible across the experiments (**Editorial Fig. 1a,b**). To further strengthen our findings, we have also measured the levels of pAkt by intracellular flow cytometry. The results shown in **Editorial Fig. 1c,d** are strikingly consistent with our western blot data, which indicate that pAkt is about 1.5-fold increase in miR-155 T cells.

Editorial Fig. 1. miR-155 enhances pAkt signaling **a** pAkt induction relative to Gapdh in Ctrl-miR or miR-155 overexpressing cells assessed by immunoblot in *in vitro* activated cells. Data representation of four independent experiments. **b** Fold increase of pAkt in miR-155 overexpressing cells relative to Ctrl-miR, shown in **a**. **c** Histograms of pAkt levels of *in vitro* activated Ctrl-miR or miR-155 overexpressing cells assessed by intracellular flow cytometry. **d** Mean fluorescence intensity (MFI) of pAkt in *in vitro* activated Ctrl-miR or miR-155 overexpressing cells.

Reviewer #2

1. With respect to minor comment 1: The authors explain their GSEA depiction based off the rank ordered list. Their interpretation in the text is as follows "Specifically, miR-155-overexpressing cells showed reduced expression of genes silenced by PRC2 in mouse and human embryonic stem cells (ESC) and progenitors^{18, 19} (Fig. 1a)". While I agree with the statement reflecting the data it would be significantly more intuitive to present the data by a heatmap rather than a GSEA."

At the reviewer request we have now provided the heatmaps of PRC2-related genes in **Supplementary Fig.1a**.

We hope you find our paper acceptable for publication in *Nature Communications*.

Sincerely,